# ImagineBench: Evaluating Reinforcement Learning with Large Language Model Rollout

## Abstract

A central challenge in reinforcement learning (RL) is its dependence on extensive real-world interaction data to learn policies. While recent work demonstrates that large language models (LLMs) can help mitigate this limitation by generating synthetic experience (noted as *imaginary rollouts*) for learning novel tasks, this area is hindered by the absence of a standardized benchmark. To bridge this gap, we propose ImagineBench, the first comprehensive benchmark for evaluating offline RL algorithms that learn from both real rollouts and LLM-imaginary rollouts. The key features of ImagineBench include: (1) datasets comprising environment-collected and LLM-imaginary rollouts with verified quality; (2) diverse domains covering locomotion, robotic manipulation, and navigation tasks; and (3) natural language task instructions of varying complexity to support instruction-following policy learning. Through comprehensive experiments, we find that simply applying existing offline RL algorithms yields suboptimal generalization on unseen tasks, achieving only 35.44% task completion on unseen tasks compared to 64.37% for policies trained with real data. Meanwhile, the performance varies with instruction complexity, confirming that ImagineBench provides meaningful spectrum of task difficulty. Furthermore, we show that pre-training with imaginary rollouts leads to superior asymptotic performance after online fine-tuning. Based on these findings, ImagineBench identifies key directions for future research, including improved exploitation of imaginary rollouts, efficient online adaptation, continual learning, and extension to multi-modal task settings. Our code is available at https://anonymous.4open.science/r/Imagine_Bench_anonymous-40CD.

## 1 Introduction

Developing knowledgeable agents that can generalize to diverse, unseen tasks represents a critical frontier in artificial intelligence. While reinforcement learning (RL) provides a framework for skill acquisition (Silver et al., 2016; Mnih et al., 2015; Vinyals et al., 2019), its reliance on extensive real-world interaction data constitutes a fundamental bottleneck for generalizing to novel tasks. In contrast, humans efficiently acquire and rehearse new skills through mental imagination, without direct physical interaction. Inspired by this capability, recent research has explored *using Large Language Models (LLMs) to generate synthetic experience, referred to as imaginary rollouts, for learning novel tasks* (Pang et al., 2024; Chen et al., 2024). This emerging paradigm, which we formalize as Reinforcement Learning from Imaginary Rollouts (RLIM), involves fine-tuning an LLM on existing environment data and then prompting it to generate synthetic rollouts for new tasks (see Fig. 1), thereby eliminating the need for initial costly interactions.

Though learning from imaginary rollouts achieves preliminary successes in robotics manipulation (Pang et al., 2024; Glossop et al., 2025), football playing (Chen et al., 2024), and browser automation (Xu et al., 2025), progress in this area is hindered by the absence of a standardized evaluation. Existing studies often employ custom environments and varied LLM architectures for algorithm evaluation. The reported performance improvements may not reliably reflect their ability to effectively utilize the foundation models' knowledge due to inconsistent evaluation protocols. Furthermore, the computational cost of fine-tuning LLMs for each new study presents a barrier to entry for many researchers, slowing progress in the field.

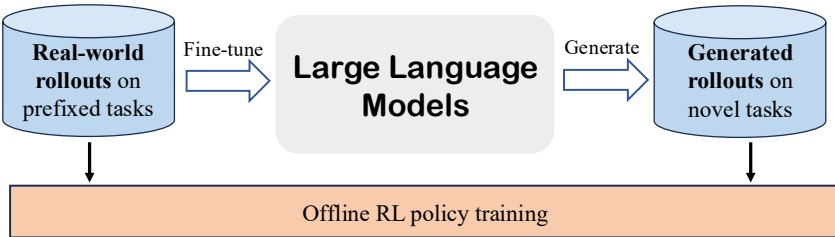

Figure 1: We benchmark the problem of RL with LLM-imaginary rollouts. The LLM is fine-tuned to generate imaginary rollouts, followed by RL policy training using real and imaginary rollouts.

To address this gap, we introduce ImagineBench, the first comprehensive benchmark designed to systematically evaluate offline RL algorithms that train a policy with both real rollouts and LLM-imaginary rollouts. ImagineBench has three key features: (1) **Datasets** that include both real rollouts collected from the environment, and imaginary rollouts generated by the fine-tuned LLMs, eliminating the computational burden of LLM fine-tuning and ensuring consistent comparison. The quality of the LLM-generated rollouts is verified by the (2) **Diverse domains** include locomotion, robotic manipulation, and navigation. (3) **Natural language instruction** paired with the rollouts, which are divided into various difficulty levels, supporting the research on instruction-following agents (Pang et al., 2023b; Ichter et al., 2022). Through extensive experiments with state-of-the-art offline RL algorithms, we demonstrate that while naively combining real and imaginary rollouts generally improves performance on unseen tasks, there is still a clear gap on novel tasks, between the current score (35.44%) and the performance of training with real rollouts (64.37%). This gap underscores the need for novel algorithms to leverage LLM-generated rollouts better. Furthermore, we show that pre-training with imaginary rollouts can enhance asymptotic performance after online fine-tuning, highlighting its potential as a valuable resource.

Our contributions are as follows: We propose ImagineBench, the first benchmark for RL from LLM-imaginary rollouts, complete with datasets, environments, and evaluation protocols. Based on ImagineBench, we conduct comprehensive empirical study investigating baseline performance and revealing the limitations of existing methods. Finally, we identify directions for future research for RL from imaginary rollouts, including improved offline RL for synthetic data, efficient online adaptation, continual learning, and extension to multi-modal tasks.

## 2 RELATED WORK

**RL with LLM-imaginary rollouts.** Recent advances in leveraging the general knowledge of LLMs to build knowledgeable agents for interactive and physical tasks have established a promising research frontier (Pang et al., 2024). The central challenge is that LLMs can not directly handle numerical control signals for decision-making tasks (Pang et al., 2024; Liu et al., 2024). To address this, researchers have explored using LLMs to generate imaginary decision-making rollouts that are then used for RL policy training. For instance, KALM (Pang et al., 2024) fine-tunes LLMs to produce low-level control rollouts, which are then used to train RL policies via offline RL algorithms. This approach demonstrates how domain-specific knowledge embedded in LLMs can be effectively distilled to handle novel tasks. Similarly, URI (Chen et al., 2024) employs LLMs to generate control trajectories by prompting them with instructional texts from tutorial books, enabling policy training without environmental interaction. AgentTrek (Xu et al., 2025) extends this paradigm to browser automation by synthesizing task execution rollouts at scale, followed by imitation learning to train the agent. Beyond low-level control, InCLET (Wang et al., 2025) introduces a framework where LLMs generate textual imaginary rollouts, enhancing the agent's ability to interpret natural language instructions and derive task representations. While these studies highlight the potential of LLM-imaginary rollouts, they focus on developing individual algorithms. In contrast, ImagineBench introduces the comprehensive benchmark to systematically evaluate the algorithm performance, generalizability, and limitations of RL methods training LLM-imaginary rollouts.

**Existing benchmarks in RL**. The rapid development of RL has given rise to a diverse array of benchmarks. These benchmarks fall into three primary categories: online, offline, and off-dynamics, each handling challenges within specific training paradigms. Online training benchmarks, such as

Figure 2: Overview of ImagineBench, covering three key features: (1) datasets of both real and LLM-imaginary rollouts, (2) diverse domains of environments, and (3) natural language instructions with various task levels. Examples shown in the 'Datasets' panel are from the CLEVR-Robot environment.

Gym (Brockman et al., 2016), MuJoCo (Todorov et al., 2012), and the DMC (Tassa et al., 2018), have long served as foundational tools for evaluating agents that learn through online interaction, emphasizing exploration and sample efficiency in dynamic settings like Atari 2600 games (Atari, Inc., 1977) and continuous control tasks. Meanwhile, the rise of offline RL promotes the development of benchmarks like NeoRL (Qin et al., 2022), D4RL (Fu et al., 2020) and RL Unplugged (Gulcehre et al., 2020), which contain large-scale, pre-collected datasets to evaluate agents' ability to learn from static data while mitigating distributional shift and extrapolation errors in domains ranging from robotic manipulation to locomotion. Besides, off-dynamics benchmarks, including ODRL (Lyu et al., 2024) and Meta-World ML1 (Yu et al., 2019), evaluate generalization under shifts in dynamics, such as altered physical parameters or visual perturbations, challenging agents to adapt policies to unseen environmental conditions. In contrast, ImagineBench is the first benchmark specifically designed to evaluate how effectively RL algorithms that utilize LLM-imaginary rollouts, offering scenarios to measure the benefits and limitations of utilizing LLM knowledge to build knowledgeable agents.

## 3 BACKGROUND

**Reinforcement learning.** We consider an RL problem where the agent completes natural language instructions. The environment can be modeled as a goal-augmented Markov Decision Process (Sutton & Barto, 1998; Pang et al., 2023a), represented by the tuple $\mathcal{M} = (\mathcal{S}, \mathcal{A}, \mathcal{P}, \mathcal{R}, \gamma, \mathcal{G})$, where $\mathcal{S}$, $\mathcal{A}$ denote the state space and action space, respectively. $\mathcal{P}$ denotes transition function of the environment, $\mathcal{R}$ the reward function that evaluates the agent's behavior, $\gamma$ the discount factor, and $\mathcal{G}$ the set of natural language goals. The objective of RL is to find a policy $\pi : \mathcal{S} \times \mathcal{G} \to \Delta(\mathcal{A})$ that maximize the cumulative reward: $J(\pi) = \mathbb{E}_\pi[\sum_{t=0}^{\infty} \gamma^t r(s_t, a_t)]$. This work focuses on environments with structured, vectorized state spaces, where each dimension encodes interpretable, domain-specific features. We call the state and action data collected from the environment the *real environmental rollouts*, and the rollouts generated by LLM the *imaginary rollouts*.

**Offline reinforcement learning with LLM-imaginary rollouts.** Traditional offline RL focuses on offline policy training from a static environmental dataset. In this paper, we consider RL with both real and LLM-imaginary rollouts. Formally, consider we have (1) a real dataset $\mathcal{D}$ collected from the real environment, and (2) a LLM-imaginary datasets[1] $\mathcal{D}^I$, which is generated by LLMs. Both real and imaginary datasets consist of paired language goals and corresponding decision-making rollouts: $\{G^k, (s_0^k, a_0^k, s_1^k, a_1^k, \cdots)\}_{k=1}^K$. Here, the sequence $(s_0^k, a_0^k, s_1^k, a_1^k, \cdots)$ represents a rollout of states and actions $(s_i^k, a_i^k)$ to complete the goal $G^k$. The primary objective is to find a policy that achieves high rewards on unseen goal distributions (known as novel tasks), represented as $\mathcal{G}'$.

## 4 IMAGINEBENCH DETAILS

ImagineBench involves a wide range of decision-making environments, including locomotion, manipulation, and navigation. For each environment, ImagineBench provides two datasets as illustrated in Sec. 3: a dataset of real rollouts collected from the environments, and a dataset of imaginary rollouts

---

[1]We will elaborate on how LLMs are trained to generate the rollouts in Sec. 4.2.

generated by LLM. We will briefly introduce the benchmark environments in Sec. 4.1 and how the datasets are constructed in Sec. 4.2. Last, Sec. 4.3 defines different levels of task complexity.

## 4.1 BENCHMARK ENVIRONMENTS

The Benchmark Environment panel in Fig. 2 shows the visualization of the environments used in ImagineBench. We present the environment statistics in Tab. 1, and more details in Appendix C.

**Meta-world** (Yu et al., 2019) agent controls a Sawyer robot to manipulate objects, e.g., doors, drawers, and windows. In novel tasks, the agent needs to manipulate, assuming that there is a wall in front of the object. The state space is $\mathbb{R}^{91}$, encoding the robot's joint angles and object positions/orientations, while the action space is $\mathbb{R}^4$, controlling the gripper's movement and open/close. The reward function combines task (or sub-task) completion signals with a negative distance metric between the gripper and target location.

**CLEVR-Robot** (Research, 2019) environment requires the agent to manipulate five colored balls to reach a target configuration. The state space is $\mathbb{R}^{10}$, encoding the positions of five balls, with an action space of 40-dimensional discrete actions, using one-hot vectors to specify directional movement for each ball. The reward is calculated as a reduction in distance between the current state and the target configuration compared to the previous step, adding a terminal reward for task completion.

**BabyAI** (Chevalier-Boisvert et al., 2019) is a gridworld environment, which modifies the original environment's language-conditioned navigation tasks with full observability. The state space is $\mathbb{R}^{17}$, encoding object positions (agent, keys, doors, balls) using absolute grid coordinates and RGB attributes. The action space comprises 7-dimensional discrete movement primitives (left/right/up/down) and object interactions (pickup/drop/toggle). The rewards are calculated as the shortest-path distance to the goal object, plus a sparse completion reward.

**LIBERO** (Liu et al., 2023) controls a robot arm to complete various manipulation tasks. LIBERO originally consists of four task suites, each containing 10 tasks. ImagineBench uses LIBERO-Object suite and additionally designs novel tasks such as sequential-pick-and-place. The state space is $\mathbb{R}^{44}$, representing the joint position and object position/poses, while the action space of $\mathbb{R}^7$ specifies joint angle deltas for arm movement and gripper open/close. Similar to Meta-world, we provide distance-based reward to guide the agent to reach the target object, and terminal judgment when a sub-task or the entire task is completed as the final step reward.

**MuJoCo** (Todorov et al., 2012) is a physics-based simulation platform widely used for continuous control tasks in reinforcement learning. In our case, ImagineBench uses the HalfCheetah robot. The state ($\mathbb{R}^{18}$) consists of positional values and velocities of different joints, while the action space ($\mathbb{R}^6$) represents the torques applied to 6 robot joints. The reward function combines forward velocity toward the target direction with control efficiency (minimizing joint torque costs).

| | Meta-world | CLEVR-Robot | BabyAI | LIBERO | MuJoCo |
|---|---|---|---|---|---|
| Observation space | $\mathbb{R}^{91}$ | $\mathbb{R}^{10}$ | $\mathbb{Z}^{17}$ | $\mathbb{R}^{44}$ | $\mathbb{R}^{18}$ |
| Action space | $\mathbb{R}^4$ | Discrete (40) | Discrete (7) | $\mathbb{R}^7$ | $\mathbb{R}^6$ |
| # of real rollout | 20,000 | 100,000 | 19,200 | 29,780 | 16,000 |
| # of IR (Rephrasing) | 10,000 | 5,600 | 19,200 | 12,000 | 10,000 |
| # of IR (Easy) | 8,000 | 72,400 | 18,000 | 24,000 | 6,000 |
| # of IR (Hard) | 4,000 | 1,680 | 18,000 | 1,3000 | 9,000 |

Table 1: Statistics overview of environments. '# of IR' stands for 'Number of imaginary rollout'.

## 4.2 DATASET COLLECTION

The dataset collection procedure consists of two steps: (1) *Real rollout collection* from the environment. In this step, we first obtain an expert policy that can complete the given tasks with a high success rate, and then use the expert policy to collect rollouts in the environment. Meanwhile, a rollout is labelled with a natural language instruction when collected. (2) *Imaginary rollout collection*

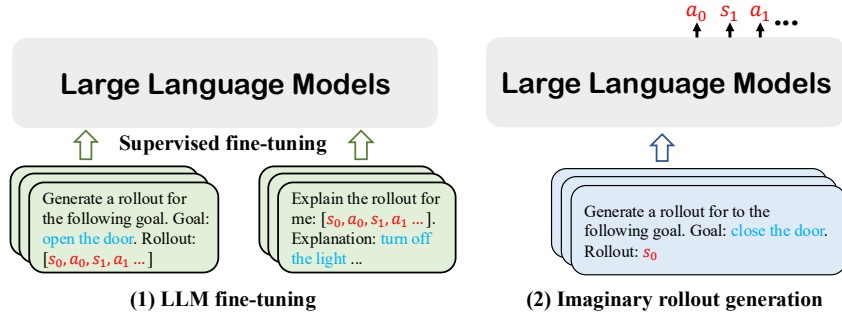

Figure 3: Illustration of the generation of LLM-imaginary rollouts. The LLM is first fine-tuned with the environment data, and then prompted to generate the rollouts for novel tasks.

from LLM. In this step, the LLM is fine-tuned on the rollout-instruction pairs from the environment, and then prompted to generate rollouts for novel tasks.

**Real rollout collection.** To collect real rollouts, we first obtain an expert policy specific to each environment and then use the policy to collect rollouts: (1) Meta-world & CLRVR-Robot: First, train an expert policy with PPO (Schulman et al., 2017), and collect an offline dataset of 20,000/100,000 rollout-goal pairs, each comprising state, action, and environment-built-in reward sequences for completing natural language goals. (2) BabyAI: Employ a rule-based policy to generate 19,200 rollout-goal pairs, with rewards based on agent-target distance. (3) LIBERO: Apply behavior cloning to public LIBERO datasets to obtain the expert policy, yielding 30,000 rollout-goal pairs with object-target distance rewards. (4) MuJoCo: Train an expert policy online using the SAC algorithm (Haarnoja et al., 2018) to collect 16,000 rollout-goal pairs. All real rollouts are annotated with natural language instructions during collection.

**Imaginary rollout collection.** Fig. 3 presents the process of fine-tuning LLM to generate imaginary rollouts[2]. To enable LLM to generate synthetic task-specific rollouts, we first fine-tune them on real rollout-instruction pairs. The objective of this step is to enable LLM to interpret the meaning of states, actions, dynamics, and rollouts of the given environment. Following (Pang et al., 2024), we fine-tune the LLM using the dataset to perform three different tasks via supervised fine-tuning (SFT), and model the LLM grounding problem as an instruction-following problem since the LLM demonstrates excellent performance following given natural language instructions to generate desired answers. The training objectives for SFT include: (1) *Dynamics prediction*: The LLM predicts changes in environmental dynamics. Given the current state $s_t$ and action $a_t$, the LLM predicts the subsequent state. (2) *Rollout explanation*: The LLM is presented with a rollout sequence $s_0, a_0, s_1, \cdots$, and it is required to describe the rollout with natural language. (3) *Rollout generation*: The LLM generates a rollout that aligns with a specified goal $G$. We present the prompts for LLM SFT in Appendix E.

Since LLMs can not directly handle numerical data, we use a pre-trained LLM as the backbone model and modify it with additional layers to handle environmental data. Then, we employ the fine-tuned LLM to generate imaginary rollouts given the initial state $s_0$ and the goal: $\{a_0, s_1, a_1, \cdots\} \leftarrow \mathcal{M}(GOP, s_0)$. Here, $\mathcal{M}$ is the LLM, $GOP$ stands for *goal-oriented prompt*: "Generate a rollout for the following goal: [GOAL]. Rollout:", where "[GOAL]" is a placeholder for various goals that reflect different skills.

**Data filtering mechanism.** In ImagineBench, we apply a minimal data selection strategy to maintain data quality, without over-filtering, to preserve the diversity of the imaginary rollouts. For real rollouts, we omit the failure trajectories that do not match their intended goals, to provide clear and reliable learning signals. For imaginary rollouts generated by the LLM, we truncate excessively long sequences to prevent episodes with redundancy that could hinder training efficiency or introduce noise. This filtering approach balances the benefits of abundant synthetic data with the need for coherent and meaningful environment interactions.

---

[2]In ImagineBench, the backbone LLMs include Qwen-3-4B-Instruct-2507 (Qwen, 2025) and Llama-2-7b-chat-hf (Touvron et al., 2023).

## 4.3 TASK HIERARCHY AND EVALUATION PROTOCOLS

ImagineBench defines hierarchical task levels indicating various levels of tasks. Due to the space constraint, we present and discuss each environment's tasks in detail in Appendix D.1.

- **Training**: The instructions appeared in the real dataset. Including training tasks is to evaluate whether the policy preserves the ability to perform these seen tasks.

- **Rephrasing**: The agent performs the same tasks as real data but receives paraphrased instructions that are not present in the data. For example, the goal in offline data is *move the blue ball to the front of the red ball*, while the paraphrased goal could be *I really dislike how the red ball is positioned in front of the blue ball. Could you exchange their places?*

- **Easy**: The agent is tasked with different manipulation tasks that do not exist in the dataset, requiring the agent to generalize to easy, unseen tasks.

- **Hard**: The agent faces tasks substantially different from those in the offline dataset, which require a complex composition of behaviors, such as "Gather all balls together", and "Move five balls to a straight line" in the CLEVR-Robot environment.

**Evaluation protocols.** We evaluate performance in ImagineBench using two primary metrics: success rate and task reward. ImagineBench provides specific success criteria for each task (e.g., achieving a specific positional accuracy in manipulation or consistent directional velocity for HalfCheetah). For detailed definitions of the completion criteria, please refer to Appendix D.3. Furthermore, each task is equipped with a designated reward function.

## 5 EXPERIMENT

In this section, we conduct experiments to address three key questions regarding ImagineBench: (1) How do existing offline RL methods perform on the tasks of ImagineBench (Sec. 5.2)? (2) For novel tasks, how does training with imaginary rollouts compare to training with real environment-collected rollouts (Sec. 5.2)? (3) How is the quality of the LLM-imaginary rollouts (Sec. 5.3)? (4) Can imaginary rollouts facilitate online adaptation (Sec. 5.4)? We first introduce the experimental setting.

### 5.1 EXPERIMENT SETTING

**Baselines.** We consider representative offline RL methods, including: (1) **BC**, a supervised learning baseline that directly imitates actions from the dataset. (2) **CQL** (Kumar et al., 2020), which learns a conservative Q-function to prevent the policy from overestimating expected returns. (3) **BCQ** (Fujimoto et al., 2019), which employs perturbation networks to generate conservative policy updates near offline data. (4) **TD3+BC** (Fujimoto & Gu, 2021), which combines TD3's ((Fujimoto et al., 2018)) stability with BC constraints to enforce similarity to demonstrated behavior. (5) **PRDC** (Ran et al., 2023), which uses a tree-search method to regularize the policy toward the nearest state-action pairs in the offline data. (6) **COMBO** (Yu et al., 2021), which uses ensemble environment models to enforce uncertainty-aware policy learning. (7) **SAC** (Haarnoja et al., 2018), which is originally an online RL algorithm, can be applied in the offline setting for comparison.

Due to the varying application scope of different algorithms, we evaluate algorithms (BC, CQL, BCQ, TD3+BC, PRDC, COMBO) on MuJoCo, LIBERO, and Meta-world, and algorithms (BC, BCQ, CQL, SAC) on CLEVR-Robot and BabyAI. *'w/ IR' represents the methods trained with both real and imaginary rollouts,* while 'w/o IR' represents methods trained solely on real rollouts.

**Implementation details.** All offline RL methods are implemented based on OfflineRL (Team, 2021) and d3rlpy (Seno & Imai, 2022), two well-established repositories. Policy optimization relies on the Adam optimizer (Kingma & Ba, 2015). Performance metrics are averaged across results from the final five training checkpoints. Unless otherwise specified, baselines encode natural language instructions using BERT (Devlin et al., 2019), and concatenate the language encoding with the environment observation. Offline RL training employs three random seeds to validate robustness. Each training batch uniformly samples equal proportions of data from the real and LLM-imaginary datasets. All experiments are executed on 64 AMD EPYC 9374F 32-core processors, 8 NVIDIA GeForce RTX 4090 GPUs, and 1TB of RAM to facilitate parallelized computation.

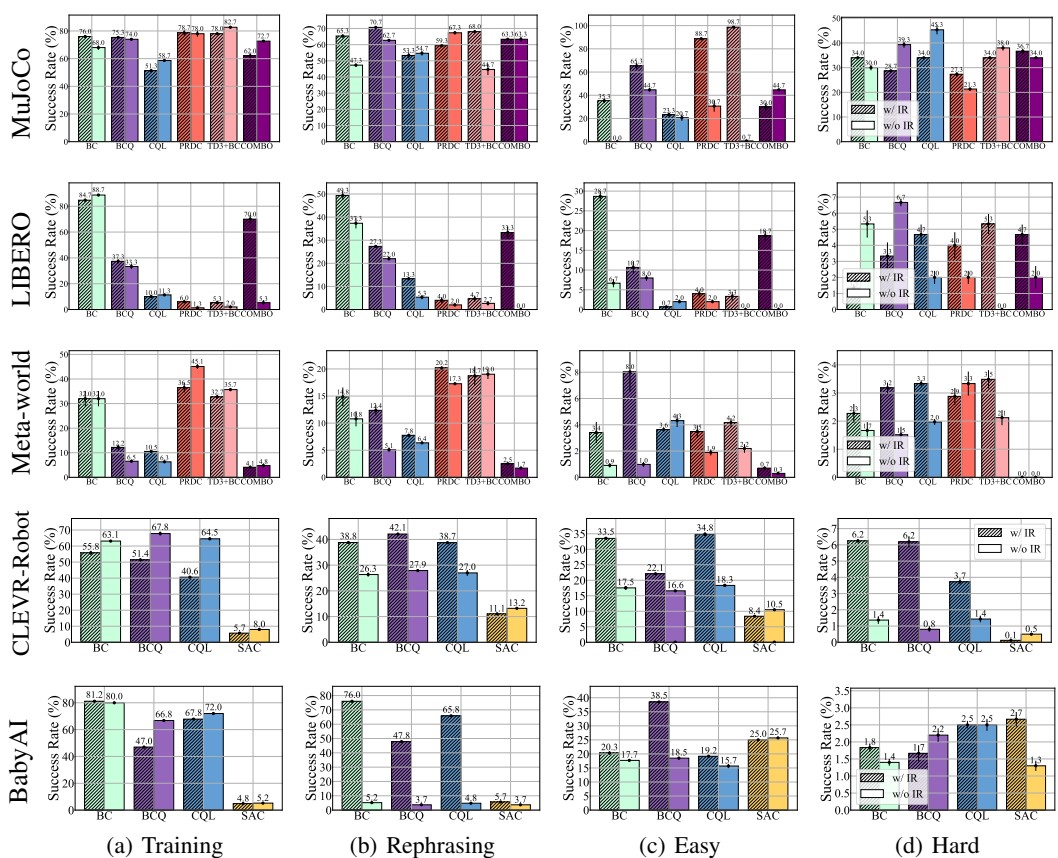

Figure 4: Success rate bars of different methods on various levels of goals, with imaginary rollouts generated by Qwen-3-4B. The x-axis denotes the offline RL algorithm, and the y-axis denotes the success rate. 'w/ IR' stands for training with both real and imaginary rollouts. The success rate is averaged over the last five checkpoints, and the error bars are the half standard deviation over three seeds. We provide the overall comparison and results for Llama-2-7B in Appendix F.2 and F.3.

## 5.2 BENCHMARK RESULTS

**Main results.** Fig. 4 presents the benchmark results of various offline RL algorithms trained with and without imaginary rollouts on ImagineBench tasks. We have several main findings from the results. First, policies trained with imaginary rollouts generally perform better on novel tasks than baseline methods. This suggests that LLM-based knowledge transfer enhances generalization and skill acquisition in unseen environments. Besides, BC, CQL, and BCQ outperform other methods across most tasks. BCQ and CQL achieve superior sample efficiency and stability in high-dimensional action spaces. As SAC is mainly used in online RL, it fails to obtain high scores in the offline cases. There is clear performance degradation on hard tasks, with most methods' success rates below 10% on Meta-World, CLEVR-Robot, and BabyAI. This gap could stem from the suboptimal reward function with current LLM rollouts, which may fail to encode task-specific constraints or long-horizon dependencies. All algorithms struggle with novel tasks on LIBERO due to its combinatorial complexity, indicating a need for advanced exploration strategies or hierarchical representations.

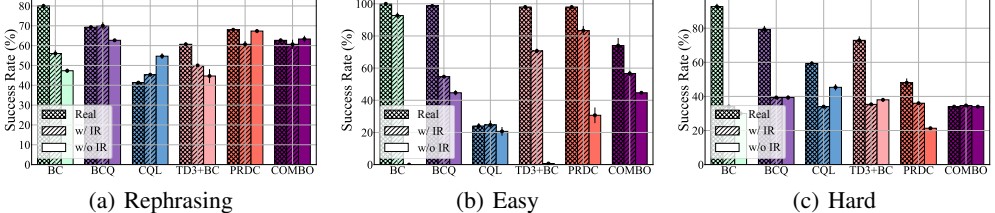

Figure 5: Comparison of training with LLM-imaginary and real environmental rollouts on novel tasks. 'Real' stands for the method trained with real environmental rollouts for novel tasks.

(a) **Success case.** Goal: *Employ the gripper to seize the salad_dressing and transfer the salad_dressing to the basket.*

(b) **Failure case.** Goal: *Employ the gripper to seize the alphabet_soup and transfer the alphabet_soup to the basket. Then employ the gripper to seize the salad_dressing and transfer the salad_dressing to the basket*

Figure 6: Examples of the LLM-imaginary rollouts for novel goals. The figures are obtained by rendering the states in LLM-imaginary rollouts. We present more examples in Appendix F.1.

**Performance of training with real rollouts on novel tasks.** To investigate the improvement space for future algorithm development, we conduct experiments by training a policy with real rollouts on both training and novel tasks. Fig. 5 shows the experiment results, with 'Real' as the method trained on real rollouts of both training and novel tasks. In most tasks, Real outperforms or gets close to the methods trained with IR, resulting in 64.37% average success rate for the Real method and 35.44% for methods with IR in hard tasks. One exception is CQL on the rephrasing task. This is because the execution rollouts of the rephrasing task have already existed the dataset of real rollouts, with only the language expression of the instructions different. The conservative learning nature of CQL allows it to focus on the state's features, potentially enabling it to perform well on rephrasing even when using only real rollouts for training tasks.

| Model | Qwen-3-4B | | | Llama-2-7B | | |
|---|---|---|---|---|---|---|
| Metrics | Legality | Transition | Success rate | Legality | Transition | Success rate |
| Rephrasing | 95.9 | 79.2 | 89.9 | 98.5 | 96.0 | 88.0 |
| Easy | 73.4 | 69.3 | 43.1 | 81.1 | 82.2 | 43.8 |
| Hard | 59.3 | 44.2 | 13.5 | 66.8 | 72.9 | 25.8 |

Table 2: Statistical analysis of the quality of LLM-imaginary rollouts. The reported results are the LLM-imaginary rollouts for the BabyAI environment.

### 5.3 ANALYSIS ON LLM-IMAGINARY ROLLOUTS

We investigate the quality of the LLM-imaginary rollouts from four key metrics: (1) *Transition* measures whether the LLM generates correct single-step transitions (e.g., an agent not moving too far at one step); (2) *Legality* denotes if the generated states are legal (i.e., the states are ); (3) *Success rate* measures the ratios of the imaginary rollouts that successfully complete the given goals. Tab. 2 reports the quality metrics of LLM-imaginary rollouts generated in the BabyAI environment. Notably, we observe an important result that larger backbone LLM (Llama-2-7B)'s generation quality clearly outperforms the small model (Qwen-3-4B). **This indicates a promising motivation that future work could investigate using larger model for better LLM imagination.** Besides, rephrasing goals achieve high-quality rollouts, with success rate, transition correctness, and legality scores of 88.0%, 96.0%, and 98.5%, respectively. This suggests that the LLM, fine-tuned on prefixed goals, generalizes effectively to semantically equivalent objectives. For novel (Hard) goals, consistency drops to 25.8%, reflecting challenges in aligning rollouts with unseen task descriptions. However, transition correctness (72.9%) and state legality (66.8%) remain above 65%, indicating that the LLM largely adheres to environmental constraints even for complex goals.

**Examples of the LLM-imaginary rollouts.** Previously we investigate the quality of the imaginary rollouts through statistics. To further investigate the quality of the generated rollouts, we present examples of the imaginary rollouts in Fig. 6. We reset the environment to the generated state to obtain the visualization image. We observe that the generated rollouts can generally reflect the given goals. For example from the success case, the robot conducts the object manipulation as the language

goal required. However, there are still some mismatches when the the goal is complicated (e.g., first pick A then pick B), where the LLM may generate wrong rollouts (e.g., simultaneous picking instead of sequential execution, as shown in the failure case). Even so, the LLM-generated rollout catches the meaning of the novel goal, and correctly demonstrates the tendency to pick up two objects.

### 5.4 Potential for Online Adaptation

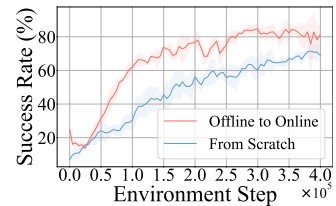

We suggest that online adaptation is the next step after training with imaginary rollouts, as the policies trained with imaginary rollouts may not be adequate for real-world deployment. To test this, we initialize a CQL policy (on CLEVR-Robot) with imaginary rollouts and then train it with PPO (Schulman et al., 2017) on Easy-level tasks. Fig. 7 shows that offline-to-online training improves adaptation speed and achieves higher asymptotic performance than online training from scratch. This demonstrates that policies trained with LLM-imaginary rollouts provide strong initialization for online adaptation.

Figure 7: Performance of training with online RL.

## 6 Future Direction

While demonstrating promising results for acquiring novel skills without online environment interactions, RL with imaginary rollouts is still in the early stage of research and requires algorithmic development. We outline key directions for future research.

**Better algorithm design for generating & utilizing imaginary rollouts.** ImagineBench reveals a performance gap between policies trained on real versus imaginary experience, which demonstrates that simply applying a powerful LLM with a sophisticated offline RL algorithm is insufficient. Future work could focus on better algorithm design to generate and handle these imaginary rollouts. For example, it is important to enhance the quality and physical property of the LLM's generative process, transforming raw imagination into high-fidelity data. Additionally, the community should design novel offline RL algorithms that are not merely consumers of this data but are specifically tailored to its unique statistical properties, including its potential for bias, noise, and distributional shift.

**Unbiased and fast online adaptation and continual learning.** While RLIM reduces dependency on real-world interactions, practical deployment still requires online adaptation to address imperfections in LLM imagination. A key challenge is avoiding catastrophic forgetting of pre-trained knowledge while rapidly fine-tuning policies with limited real interactions. Future research could consider developing lightweight regularization techniques to preserve imaginary knowledge, meta-RL frameworks for few-shot adaptation, or progressive distillation methods to compress multi-task policies. Furthermore, designing bias correction mechanisms to disentangle inaccuracies in LLM-generated rollouts during online updates could enhance sample efficiency and stability.

**Vision-Language Models and Multi-Modal Imagination.** Current benchmark mainly focuses on the environment state represented by structural and numerical vectors. Extending RLIM to broader domains, e.g., vision, requires integrating vision-language models capable of processing and generating multi-modal rollouts. This entails addressing challenges such as aligning visual observations with language instructions, generating spatially consistent action sequences from pixel inputs, and handling partial observability in imagined states. Future work could explore cross-modal attention mechanisms for joint rollout generation or develop hierarchical frameworks where high-level language plans guide low-level visual motion generation.

## 7 Conclusion

In this work, we present ImagineBench, the first benchmark for RL with LLM-imaginary rollouts. By providing standardized datasets across locomotion, robotic manipulation, and navigation environments, ImagineBench establishes a unified framework to evaluate offline RL algorithms that utilize the LLM-imaginary rollouts. The benchmark results reveal the limitations of existing offline RL methods when applied to LLM-imaginary datasets, underscoring the necessity for algorithmic innovations that better integrate LLM-generated knowledge. Beyond benchmarking, ImagineBench is a resource to advance the development of agents that can not only execute predefined tasks but also generalize to unseen ones, marking a foundational step toward robust embodied intelligence.

## 8 ETHICS STATEMENT

This work adheres to the ICLR Code of Ethics and prioritizes responsible research practices. All natural language instructions used in ImagineBench are carefully curated and sanitized to exclude harmful, biased, or ethically problematic content, using both automated filtering and manual expert review. The LLM-generated imaginary rollouts are released exclusively as numerical state-action sequences—not as human-readable plans or executable code—to inherently limit potential misuse. Our benchmark is built entirely on simulated environments (e.g., MuJoCo, Meta-World, BabyAI), contains no human-subject data, and does not involve real-world deployment or personal information. By design, ImagineBench supports open, reproducible research while incorporating structural safeguards to align with principles of fairness, transparency, and societal benefit.

## 9 REPRODUCIBILITY STATEMENT

To support reproducibility, we provide a comprehensive anonymous codebase at https://anonymous.4open.science/r/Imagine_Bench_anonymous-40CD, which includes implementations of all environments, dataset loaders, offline RL baselines, and evaluation protocols used in this work. Detailed instructions for reproducing our main results are given in Appendix E, including environment setup and training commands. The full datasets of real and LLM-imaginary rollouts, along with task definitions and natural language instructions, are included in the supplementary materials; the download link has been omitted to preserve anonymity during double-blind review but will be made publicly available upon acceptance.

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

# Appendix

## Table of Contents

## A   ADDITIONAL RELATED WORK ABOUT OFFLINE RL

This work considers utilizing offline RL algorithms to train the policy. Offline RL (Levine et al., 2020; Fujimoto et al., 2019) enables agents to learn effective policies from static datasets without online environment interactions. Early approaches to offline RL, such as BCQ (Fujimoto et al., 2019) and BEAR (Kumar et al., 2019), addressed distributional shift by constraining learned policies to remain close to the behavior policy through explicit policy regularization or uncertainty-based action clipping. Subsequent advances introduced CQL (Kumar et al., 2020), which penalizes Q-value overestimation for out-of-distribution actions, and implicit constraint methods like TD3+BC (Fujimoto & Gu, 2021) that balance policy improvement with behavior cloning. Decision transformer (Chen et al., 2021) has also explored leveraging trajectory-level optimization via sequence modelling. Despite these advancements, offline RL remains constrained by dataset quality: policies trained on narrow or non-diverse data often fail in unseen scenarios. Model-based RL (Luo et al., 2024) addresses this by learning a dynamics model from offline data, enabling policy optimization through simulated rollouts. Methods like MOPO (Yu et al., 2020) and MOReL (Kidambi et al., 2020) incorporate uncertainty quantification to construct pessimistic models, mitigating model bias and distributional mismatch. In this work, we utilize offline RL methods to train the policy, providing the benchmark results.

## B   MORE DISCUSSIONS ABOUT RL WITH IMAGINARY ROLLOUTS

**Potential of RL with imaginary rollouts.** Generalization is a long-standing problem in the area of RL. The motivation of developing RLIM algorithms, is to leverage the general knowledge embedded in the LLMs to facilitate RL policy's generalization to unseen decision-making task. Previously RL lacks such general knowledge for generalization. This way of using imaginary rollouts imitates the human process of acquiring novel skills, i.e., imagining the process of the new objective, and executing following the imagination. However, the quality of imaginary rollouts remain to be improved, e.g., incorporating advancements in generative artificial intelligence techniques to better align generated rollouts with novel tasks.

**Better LLM imagination.** The quality of imaginary rollouts remains a limiting factor, as current LLMs often generate rollouts inconsistent with the given instructions. To address this, it is worth considering improving LLM fine-tuning, integrating physics-based simulators to validate generated rollouts, or developing iterative imagination procedure where policy learning and LLM generation both get improvement. Additionally, scaling laws for LLM imagination, exploring how model size, model type, prompt engineering, and affect rollout quality, also require systematic investigation.

## C   MORE DETAILS ABOUT BENCHMARK ENVIRONMENTS

### C.1   STATE SPACE DECOMPOSITION AND FEATURE ATTRIBUTION

**Meta-world** This environment consists of a robotic gripper and several (2 at most) interactive objects, where the state space represents the coordinate values of both the robotic gripper and the interactive objects. The elements are in Tab. 3

**CLEVR-Robot** This environment contains five colored balls. The state space encodes the position of five balls. The elements are in Tab. 4

**BabyAI** This environment is based on the grid world scenario containing an agent and a few different objects. We use one room and place one item for all types of object. The state space represents each item's color and coordinate, together with extra information including agent position, carrying object and door state. The elements are in Tab. 5

**Libero** The Libero environment controls a 3-dimensional robot arm to complete various manipulation tasks. The state space is $\mathbb{R}^{44}$, consisting of 7 robot joint position values, 7 robot end effector position values and 2 robot gripper joint position values, together with 28 position values of 4 different objects. The elements are in Tab. 6.

**MuJoCo** We use mujoco HalfCheetah environment. This environment is a 2-dimensional robot consisting of 9 body parts and 8 joints connecting them (including two paws). The state space is $\mathbb{R}^{18}$, consisting of 9 position values and 9 velocities of the robot's body parts. The elements are in Tab. 7.

| Index | Meaning | Min | Max |
|---|---|---|---|
| 0 | current x coordinate of robotic gripper | -0.525 | 0.525 |
| 1 | current y coordinate of robotic gripper | 0.348 | 1.025 |
| 2 | current z coordinate of robotic gripper | -0.0525 | 0.7 |
| 3 | current opening degree of robotic gripper | -1.0 | 1.0 |
| 4 - 6 | current x,y,z coordinate of First interactive objects | -Inf | Inf |
| 7 - 10 | current quaternion(s) of First interactive objects | -Inf | Inf |
| 11 - 13 | current x,y,z coordinate of Second interactive objects | -Inf | Inf |
| 14 - 17 | current quaternion(s) of Second interactive objects | -Inf | Inf |
| 18 | previous (last step) x coordinate of robotic gripper | -0.525 | 0.525 |
| 19 | previous (last step) y coordinate of robotic gripper | 0.348 | 1.025 |
| 20 | previous (last step) z coordinate of robotic gripper | -0.0525 | 0.7 |
| 21 | previous (last step) opening degree of robotic gripper | -1.0 | 1.0 |
| 22 - 24 | previous (last step) x,y,z coordinate of First interactive objects | -Inf | Inf |
| 25 - 28 | previous (last step) quaternion(s) of First interactive objects | -Inf | Inf |
| 29 - 31 | previous (last step) x,y,z coordinate of Second interactive objects | -Inf | Inf |
| 32 - 35 | previous (last step) quaternion(s) of Second interactive objects | -Inf | Inf |
| 36 - 38 | current x,y,z coordinate of goal position | -Inf | Inf |

Table 3: State space decomposition of Meta-World environment.

| Index | Meaning | Min | Max |
|---|---|---|---|
| 0 - 1 | red ball x,y coordinate | -Inf | Inf |
| 2 - 3 | blue ball x,y coordinate | -Inf | Inf |
| 4 - 5 | green ball x,y coordinate | -Inf | Inf |
| 6 - 7 | purple ball x,y coordinate | -Inf | Inf |

Table 4: State space decomposition of CLEVR-Robot environment.

## C.2 ACTION SPACE DECOMPOSITION

**Meta-world** In Meta-World, the action space is a 2-tuple consisting of the change in 3D space of the end-effector followed by a normalized torque that the gripper fingers should applyan. The elements are in Tab.8

**CLEVR-Robot** In CLEVR-Robot, an action is pushing one certain ball to a certain direction. The elements are in Tab.9

**BabyAI** In BabyAI, actions directly controls the agent. Possible actions include move, pick up, drop and open. The elements are in Tab.10

**Libero** The Libero environment controls a 7-degree-of-freedom (DoF) PandaGripper using delta pose control. The action space is $\mathbb{R}^7$. An action represents changes in the Cartesian position and orientation of the robot, along with the gripper actuation.The elements are in Tab. 11.

**MuJoCo** The MuJoCo HalfCheetah robot's torso and head are fixed, and torque can only be applied to the other 6 joints over the front and back thighs, the shins, and the feet. The action space is $\mathbb{R}^6$. An action represents the torques applied at the hinge joints. The elements are in Tab. 12.

| Index | Meaning | Min | Max |
|---|---|---|---|
| 0 | ball color | 0 | 5 |
| 1 - 2 | ball x,y coordinate | 1 | 6 |
| 3 | box color | 0 | 5 |
| 4 - 5 | box x,y coordinate | 1 | 6 |
| 6 | key color | 0 | 5 |
| 7 - 8 | key x,y coordinate | 1 | 6 |
| 9 | door color | 0 | 5 |
| 10 - 11 | door x,y coordinate | 1 | 6 |
| 12 | door close or not | 0 | 1 |
| 13 - 14 | agent x,y coordinate | 1 | 6 |
| 15 | carrying object type | 5 | 7 |
| 16 | carrying object color | 0 | 5 |

Table 5: State space decomposition of BabyAI environment.

| Index | Meaning | Min | Max |
|---|---|---|---|
| 0 - 6 | joint position of the robot arm | -Inf | Inf |
| 7 - 9 | position of the robot end effector | -Inf | Inf |
| 10 - 13 | quaternion of the robot end effector | -1 | 1 |
| 14 - 15 | joint position of robot gripper | -Inf | Inf |
| 16 - 18 | position of alphabet_soup | -Inf | Inf |
| 19 - 22 | quaternion of alphabet_soup | -1 | 1 |
| 23 - 25 | position of cream_cheese | -Inf | Inf |
| 26 - 29 | quaternion of cream_cheese | -1 | 1 |
| 30 - 32 | position of salad_dressing | -Inf | Inf |
| 33 - 36 | quaternion of salad_dressing | -1 | 1 |
| 37 - 39 | position of basket | -Inf | Inf |
| 40 - 43 | quaternion of basket | -1 | 1 |

Table 6: State space decomposition of LIBERO environment.

# D  MORE DETAILS ABOUT HIERARCHICAL TASKS

## D.1  FULL LIST OF THE TASKS IN IMAGINEBENCH

Tab. 13 shows the task list of all tasks for each environment. We present some examples of natural language instructions for these tasks in Appendix D.4.

## D.2  REWARD DESIGN FOR EACH TASK

### D.2.1  META-WORLD

The rewards for all tasks within the Meta-world environment are determined using the original Meta-world environment rewards, supplemented by reward shaping techniques. The reward function is defined as:

$$r_t = r_{\text{success}} + (r_t^o - r_{t-1}^o)$$

| Index | Meaning | Min | Max |
|---|---|---|---|
| 0 | x-coordinate of the front tip | -Inf | Inf |
| 1 | z-coordinate of the front tip | -Inf | Inf |
| 2 | angle of the front tip | -Inf | Inf |
| 3 | angle of the back thigh | -Inf | Inf |
| 4 | angle of the back shin | -Inf | Inf |
| 5 | angle of the back foot | -Inf | Inf |
| 6 | angle of the front thigh | -Inf | Inf |
| 7 | angle of the front shin | -Inf | Inf |
| 8 | angle of the front foot | -Inf | Inf |
| 9 | velocity of the x-coordinate of front tip | -Inf | Inf |
| 10 | velocity of the z-coordinate of front tip | -Inf | Inf |
| 11 | angular velocity of the front tip | -Inf | Inf |
| 12 | angular velocity of the back thigh | -Inf | Inf |
| 13 | angular velocity of the back shin | -Inf | Inf |
| 14 | angular velocity of the back foot | -Inf | Inf |
| 15 | angular velocity of the front thigh | -Inf | Inf |
| 16 | angular velocity of the front shin | -Inf | Inf |
| 17 | angular velocity of the front foot | -Inf | Inf |

Table 7: State space decomposition of MuJoCo environment.

| Num | Action |
|---|---|
| 0 | $\Delta x$ of the robotic gripper |
| 1 | $\Delta y$ of the robotic gripper |
| 2 | $\Delta z$ of the robotic gripper |
| 3 | opening degree of robotic gripper |

Table 8: Action space decomposition of Meta-World environment.

$r_{\text{success}} \in \{10, 0\}$ indicate whether the task has been successfully completed. $r_t^o$ stands for the original Meta-world environment reward at time step $t$.

Additionally, there are two self-designed environments. In the Make-coffee task, it can be decomposed into two sub-tasks: Coffee-push and Coffee-button-press. Similarly, the Locked-door-open task can be separated into two sub-tasks: Door-unlock and Door-open. The variable $r^o$ represents the reward associated with the task being performed.

### D.2.2 CLEVR-ROBOT

For Training tasks and Rephrasing tasks, the distance-based reward function is defined as:
$$r_t = r_{\text{success}} + (d_{t-1} - d_t) * 10$$

$r_{\text{success}} \in \{100, 0\}$ indicate whether the task has been successfully completed. $d_t$ is the distance between two balls.

For Easy tasks, sparse reward is utilized because the task can be and must be accomplished in a single step. Specifically, $r_t = 1$ when the action taken is desired; otherwise, $r_t = 0$.

For Hard tasks, the reward function based on sub-goal is defined as:
$$r_t = r_{\text{success}} + (g_{t-1} - g_t) * 10$$

| Num | Action |
| --- | --- |
| 0 - 3 | push the red ball to right,back,left,front |
| 4 - 7 | push the red ball to right rear,left rear,right font,left font |
| 8 - 11 | push the blue ball to right,back,left,front |
| 12 - 15 | push the blue ball to right rear,left rear,right font,left font |
| 16 - 19 | push the green ball to right,back,left,front |
| 20 - 23 | push the green ball to right rear,left rear,right font,left font |
| 24 - 27 | push the purple ball to right,back,left,front |
| 28 - 31 | push the purple ball to right rear,left rear,right font,left font |
| 32 - 35 | push the cyan ball to right,back,left,front |
| 36 - 39 | push the cyan ball to right rear,left rear,right font,left font |

Table 9: Action space decomposition of CLEVR-Robot environment.

| Num | Action |
| --- | --- |
| 0 | move left |
| 1 | move right |
| 2 | move up |
| 3 | pick up the object in current grid |
| 4 | drop carrying object in current grid |
| 5 | open door around agent |
| 6 | move down |

Table 10: BabyAI env action space

$r_{\text{success}} \in \{10, 0\}$ indicates whether the task has been successfully completed. The variable $g_t$ represents the number of sub-goals completed at the time step $t$.

For Sequential-move, each sub task is a move task.

For Make-line, the task requires all five balls $b_1, \ldots, b_5$ are placed in a sequential horizontal alignment. A sub-task is positioning ball $b_i$ adjacent to $b_{i+1}$ horizontally.

For Make-circle, the objective is to arrange all other balls in proximity to the green ball, with each individual sub-task involving the placement of one additional ball adjacent to the green ball.

### D.2.3 BABYAI

Reward of all tasks of BabyAI is calculated based on agent-object distance. The reward function is defined as:

$$r_t = r_{\text{success}} + \frac{d_{t-1} - d_t}{d_0}$$

$r_{\text{success}} \in \{1 - 0.9 \times \frac{\text{step\_count}}{\text{max\_steps}}, 0\}$ indicate whether the task has been successfully completed. $d_t$ stands for the agent-object distance at time step $t$.

In task Goto, Pickup, Open, Go-wall, Go-center $d$ is the Manhattan Distance between agent and target object or position.

For Put-next, Open-go, Open-pick, Open-lock, the task can be divided into two sub tasks. So the distance is defined as $d = d_1 + d_2 + p$. $d_1$ is the Manhattan Distance between agent and object 1, $d_1 = 0$ if sub task 1 is accomplished. $d_2$ is the Manhattan Distance between object 1 and object 2 if sub task 1 has not been accomplished else the Manhattan Distance between agent and object 2. $p$ is penalty for not accomplishing sub task 1 and unwanted pickups.

| Num | Action |
|:---:|:---:|
| 0 | change in x-coordinate of the gripper |
| 1 | change in y-coordinate of the gripper |
| 2 | change in z-coordinate of the gripper |
| 3 | change in x-rotation of the gripper |
| 4 | change in y-rotation of the gripper |
| 5 | change in z-rotation of the gripper |
| 6 | gripper open and close control |

Table 11: Action space decomposition of LIBERO environment.

| Num | Action |
|:---:|:---:|
| 0 | torque applied on the back thigh rotor |
| 1 | torque applied on the back shin rotor |
| 2 | torque applied on the back foot rotor |
| 3 | torque applied on the front thigh rotor |
| 4 | torque applied on the front shin rotor |
| 5 | torque applied on the front foot rotor |

Table 12: Action space decomposition of MuJoCo environment.

For Put-line, Put-pile, $d$ is defined as the sum of the grid number each object need to pass through to form the shape of a line or a pile.

### D.2.4 LIBERO

Reward of all Libero tasks is based on the distance between current state and target state. The reward function is defined as:

$$r_t = r_{\text{success}} + \alpha \cdot r_{\text{distance}}$$

The term $r_{\text{success}}$ indicates whether the current task has been successfully completed. The agent receives a $r_{\text{success}}$ of +1 if it accomplishes a sub-task or the entire task. For Pick, Place and Reach tasks, the agent only receives a +1 reward if it accomplishes the entire task successfully. While for some complex manipulation tasks, such as Pick-and-place, Pick-out, Pick-and-place aside and Sequential-pick-and-place, the agent first accomplishes a sub-task and then the next. For example, in sequential-pick-and-place tasks, the agent grasps the object, places the object to the target position and then repeats the same process for the next object. In these complex tasks, the agent receives a +1 reward if the current sub-task is successfully completed for the first time.

The term $r_{\text{distance}}$ indicates the change in distance between current state and the target state, which can also be written as $d_t - d_{t+1}$. For all Libero tasks, we use Manhattan Distance to calculate distance between states. For Pick tasks and complex tasks with pick operation as current sub-task, distance is calculated by the gripper position and the target object position. While for Place tasks and complex tasks with place operation as current sub-task, distance is calculated by the object position and the target position.

The term $\alpha$ is a weighting coefficient that balance $r_{\text{success}}$ and $r_{\text{distance}}$.

### D.2.5 MUJOCO

Reward of all MuJoCo tasks is based on the forward distance in the target direction, along with control efficiency. The reward function is defined as:

$$r_t = r_{\text{forward}} + r_{\text{control}}$$

| | **Meta-world** | **CLEVR-Robot** | **BabyAI** | **LIBERO** | **MuJoCo** |
|---|---|---|---|---|---|
| Training task | Reach, Push, Pick-place, Button-press, Door-unlock, Door-open, Window-open, Faucet-open, Coffee-push, Coffee-button-press | Move | Goto, Pickup, Open, Put-next | Pick, Place | Run-forward, Run-backward, Jump-forward, Jump-backward |
| Rephrasing task | Same as training (with rephrasing instructions) | | | | |
| Easy task | Reach-wall, Push-wall, Pick-place-wall, Button-press-wall, Door-lock, Door-close, Window-close, Faucet-close | One-step-move | Open-go, Open-pick, Go-wall, Go-center | Pick-and-place, Pick-and-place-to-unseen, Reach | Run-forward-faster, Run-backward-faster |
| Hard task | Make-coffee, Locked-door-open, Hammer, Soccer | Sequential-move, Make-line, Make-circle | Open-lock, Put-line, Put-pile | Sequential-pick-and-place, Pick-and-place-aside, Pick-out | Run-forward-then-backward, Run-backward-then-forward, Jump-in-place |

Table 13: Full lists of tasks for each environment.

The term $r_{\text{forward}}$ is a reward for moving in the right direction. This term can also be written as $\omega_{\text{forward}} \cdot \frac{dx}{dt}$, where $\omega_{\text{forward}}$ is the forward reward weight (default is 1), $dx$ is the displacement of the tip in the right direction and $dt$ is the time between actions (default is 0.05).

The term $r_{\text{control}}$ is a negative reward using L2 norm of action $a_t$ to penalize the robot for taking actions that are too large. This term can also be written as $-\omega_{\text{control}}\|a_t\|_2^2$, where $\omega_{\text{control}}$ is set to 0.1 by default.

### D.3 DETERMINATION FOR TASK COMPLETION

#### D.3.1 META-WORLD

- The metrics for evaluating success based on gripper-target distance utilized in Meta-world environments are identical to those implemented in the original Meta-world environments.

- For Make-coffee and Locked-door-open, these two self-design task can be divided into two distinct sub tasks. Consequently, the task is deemed successfully completed when both sub-tasks are accomplished sequentially.

#### D.3.2 CLEVR-ROBOT

- Training tasks and Rephrasing tasks: these tasks are considered complete when the angular relationship between the two balls satisfies the specified direction (such as left or right), and the distance $d_t$ between them is less than 0.39.

- For Easy tasks, where a single-step action is required, the task is deemed successful if the desired action is chosen; otherwise failed.

- For Hard tasks, the task can be segmented into a few sub tasks. The task is considered complete if all sub tasks are completed, irrespective of the sequence in which they are completed. For Sequential-move, each sub task is a move task. For Make-line, the task requires arranging all five balls $b_1, \ldots, b_5$ in a sequential horizontal line. A specific sub-task involves aligning each pair of consecutive balls, $b_i$ and $b_{i+1}$, horizontally. For the sub-task, the angle deviation from the horizontal line should be less than $\frac{\pi}{6}$. For Make-circle, the

objective is to arrange all additional balls surrounding the green ball, where each individual task involves placing one of the other balls adjacent to the green ball. The criterion for "adjacent" is defined as having a distance $d < 0.325$ between the two balls.

### D.3.3 BABYAI

- Goto, Go-wall, Go-center: the agent reaches the desired position.
- Pickup: the target object is picked up.
- Open: the door is opened.
- Put-next, Put-line, Put-pile: the three objects form the desired shape.
- Open-go, Open-pick, Open-lock: the task can be divided into two sub tasks. So the task is considered to be complete if two sub task is completed in correct order.

### D.3.4 LIBERO

- Pick: the task is considered completed if the robot gripper is close enough to the target object and the position of the target object is changed compared with the last state. In these tasks, completion judgment can be formally written as $d_t < \epsilon$ and $\|Pos^t_{obj} - Pos^{t+1}_{obj}\|^2_2 > 0$, where $\epsilon$ varies with different objects.
- Reach: the task is considered completed if the robot gripper is close enough to the target object. For Place tasks, the task is considered completed if the object is close enough to the target position. In these tasks, completion judgment can be formally written as $d_t < \epsilon$, where $\epsilon$ varies with different objects.
- Pick-out: the task is considered completed if the object is far enough from the object's initial position. In these tasks, completion judgment can be formally written as $d_t > \epsilon$.
- For complex tasks in which the agent accomplishes different sub-tasks sequentially, the task is considered completed if each sub-task is completed in given order.

### D.3.5 MUJOCO

- Jump: the robot completes a jump operation in the right direction. The correctness of direction can be judged by the symbol of cumulative distance in x-coordinate.
- Run and Run-faster: the cumulative distance in x-coordinate exceeds a pre-defined maximum distance value. In these tasks, completion judgment can be formally written as $\sum_{i=1}^{t} d_i > d_{max}$, where $d_{max}$ varies with different tasks. For Run-faster tasks, $d_{max}$ is a larger value compared with that in Run tasks.
- Run-forward-then-backward and Run-backward-then-forward: the robot completes both run forward and run backward operations.
- Jump-in-place: the robot completes a jump operation without a large cumulative distance in x-coordinate. In this task, completion judgment can be formally written as $\sum_{i=1}^{t} d_i < d_{min}$.

### D.4 NATURAL LANGUAGE INSTRUCTIONS FOR DIFFERENT TASKS

In this section, we present the natural language instructions for all tasks in ImagineBench for readers' reference. Note that here *we only present partial natural language instructions for each task for better reading purpose*. Please check the full instruction list in our open-sourced codebase.

### D.4.1 META-WORLD

**Training** We use 20 different natural language expressions as training goals generated by ChatGPT to express different target configuration.

- Reach task
    1. Relocate the gripper to the designated spot.
    2. Position the gripper at the intended location.

- Push task
    1. Employ the gripper to propel the target object towards its designated location.
    2. Utilize the gripper to advance the target object to its intended position.
- Pick-place task
    1. Employ the gripper to seize the designated item and transfer it to the specified position.
    2. Utilize the gripper for grasping the desired object and relocating it to the designated spot.
- Button-press task
    1. Utilize the gripper to firmly depress the button.
    2. Apply pressure with the gripper to activate the button.
- Door-unlock task
    1. Employ the gripper to turn the door's unlocking mechanism.
    2. Utilize the gripper to manipulate the lock and open the door.
- Door-open task
    1. Utilize the gripper to grasp the door handle and pull it open.
    2. Employ the gripper to grip the door handle and swing it outward.
- Window-open
    1. Employ the clamping tool to pry the window open.
    2. Utilize the grabbing device for window aeration.
- Faucet-open
    1. Employ the gripping tool to turn the faucet on.
    2. Utilize the clamp to twist the tap open.
- Coffee-push
    1. Employ the gripper to nudge the coffee beneath the coffee machine.
    2. Utilize the gripper to slide the coffee under the coffee machine.
- Coffee-button-press
    1. Utilize the gripper to depress the button on the coffee machine.
    2. Employ the gripper to push down the button of the coffee machine.

**Rephrasing** We use 20 different natural language expressions as the novel goals generated by ChatGPT to express different target configuration.

- Reach task
    1. I'm dissatisfied with the gripper's current location; kindly adjust it to reach the desired position.
    2. The gripper's current placement doesn't suit me; could you relocate it to the target position?
- Push task
    1. The current location of the target object isn't satisfactory to me; please utilize the gripper to nudge it to the target position.
    2. I'm not pleased with where the target object is currently situated; could you employ the gripper to guide it to the intended position?
- Pick-place task
    1. I have a negative sentiment towards the current placement of the object of interest; therefore, I intend to utilize the gripper mechanism to lift it and relocate it to the desired destination.
    2. The current arrangement of the designated item is unsatisfactory to me, prompting me to employ the gripper for the purpose of relocating it to the specified destination.
- Button-press task

1. I have a displeasure towards the inactive state of the button; therefore, I intend to utilize the gripper to apply pressure and activate it in order to open it.
2. The current state of the button being inactive is not to my liking, prompting me to use the gripper to press it and initiate its function of opening.

- Door-unlock task

    1. I despise when the door is locked; could you employ the gripper to unlock it?
    2. I loathe it when the door is locked; kindly utilize the gripper to release it?

- Door-open task

    1. I detest when the door is closed; could you utilize the gripper to open it, please?
    2. I can't stand it when the door is closed; kindly employ the gripper to open it for me?

- Window-open task

    1. I dislike it when the window is shut; could you kindly employ the gripper to unlatch it?
    2. I have a strong aversion to the closed window; would you mind utilizing the gripper to open it?

- Faucet-open task

    1. I dislike it when the faucet is shut; could you kindly utilize the gripper to turn it on?
    2. I have a strong aversion to the closed faucet; would you mind employing the gripper to open it?

- Coffee-push task

    1. I despise the coffee's current location; utilize the gripper to shift it to the desired spot.
    2. The coffee's present placement irks me; employ the gripper to relocate it to its intended position.

- Coffee-button-press task

    1. I believe the coffee machine shouldn't be switched off; utilize the gripper to press its button and activate it.
    2. I disagree with the coffee machine being off; employ the gripper to push its button and power it up.

**Easy** We use 20 different natural language expressions as the novel goals generated by ChatGPT to express different target configuration. Natural language instruction can be one of the following:

- Reach-wall task

    1. Adjust the gripper's position to reach the designated target, keeping in mind the obstructing wall.
    2. Maneuver the gripper towards the desired location, taking into consideration the presence of a barrier.

- Push-wall task

    1. Employ the gripper to propel the target object towards the designated location, noting the nearby wall obstructing the path.
    2. Utilize the gripper to push the target object towards its destination, recognizing the presence of a wall blocking the middle of the path.

- Pick-place-wall task

    1. Utilize the gripper apparatus to grasp the designated object and transfer it to the intended position, notwithstanding the obstruction posed by a wall at the target site.
    2. Employ the gripper mechanism to seize the desired item and relocate it to the specified spot, recognizing the hindrance presented by a wall obstructing the target destination.

- Button-press-wall task

    1. Employ the gripper to depress the button, yet a wall has emerged, obstructing access.
    2. Utilize the gripper for pushing the button, only to encounter an impediment in the form of a wall.

- Door-lock task
    1. Utilize the gripper to secure the door shut.
    2. Employ the gripper to fasten the door securely.
- Door-close task
    1. Employ the gripper to shut the door.
    2. Utilize the gripper to seal the door.
- Window-close task
    1. Utilize the gripper to shut the window.
    2. Employ the gripper to seal the window.
- Faucet-close task
    1. Utilize the gripper to shut off the faucet.
    2. Employ the gripper to seal the faucet.

**Hard** We use 20 different natural language expressions as the novel goals generated by ChatGPT to express different target configuration. Natural language instruction can be one of the following:

- Make-coffee task
    1. Utilize the gripper to position the coffee mug beneath the coffee machine nozzle, ensuring proper alignment.
    2. Employ the gripper mechanism to slide the coffee cup into place beneath the coffee machine's dispenser.
- Locked-door-open task
    1. Would you kindly unlock and open the door using the gripper?
    2. Please utilize the gripper to unlock and then open the door.
- Hammer task
    1. Utilize the gripper to grasp the hammer and strike the nail at the designated spot.
    2. Employ the gripper for seizing the hammer and driving the nail into the target location.
- Soccer task
    1. Utilize the gripper to propel the football into the goal at the designated spot.
    2. Employ the gripper mechanism to push the football into the goal at the specified location.

### D.4.2 CLEVR-ROBOT

**Training/Rephrasing** We use 40 different natural language expressions as the novel goals generated by ChatGPT to express different target configuration. For example, if we take a goal configuration such as "red ball and blue ball", its corresponding natural language instruction can be one of the following:

- I can't stand the red ball ahead of the blue one. Could you switch the positions of them?
- The sight of the red ball ahead of the blue one bothers me. Can we reverse their order?
- I really dislike how the red ball is positioned in front of the blue ball. Could you exchange their places?
- It annoys me to see the red ball in front of the blue ball. Can we swap them around?
- Seeing the red ball ahead of the blue ball fills me with frustration. Let's switch them.
- The placement of the red ball in front of the blue ball is something I detest. Can you flip them?

**Easy** In easy task, the agent needs to move one ball to a specific direction. The natural language goal can be one of the following:

- Move the ball backward, it's red.
- Push the red ball in reverse.
- Back up the red ball, please.
- Shift the red ball backwards.
- Can you move the red ball backwards?
- Retract the red ball, moving it backwards.

**Hard** We designed 4 types of completed unseen tasks: combination of two simple tasks, combination of three simple tasks, object arrangement task, and object collection task.

- Natural language sentence patterns used in combination of simple tasks (Using "red ball *behind* blue ball" as goal configuration):
    1. Push the red ball behind the blue ball.
    2. Move the red ball behind the blue ball.
- Combination of two simple tasks: Push the red ball behind the blue ball and move the green ball behind the purple ball.
- Combination of three simple tasks: Push the red ball behind the blue ball and move the green ball to the left of the purple ball and keep the cyan ball in front of the red ball.
- Object arrangement task
    1. Place the balls horizontally, lining them up from left to right, in the order of red, blue, green, purple, and cyan.
    2. Set up the balls in a row from left to right, with red, blue, green, purple, and cyan in sequence.
- Object collection task
    1. Position all the other balls around the green ball, considering it as the circle's focal point.
    2. Use the green ball as the nucleus of the circle, arranging the rest around it.

### D.4.3 BABYAI

**Training** We use 40 different natural language expressions as training goals generated by ChatGPT to express different target configuration. For example, if we take a goal configuration such as "red ball, blue key, green door", its corresponding natural language instruction can be one of the following:

- Goto task
    1. go to the red ball.
    2. move to the red ball.
    3. head toward the red ball.
    4. walk to the red ball.
    5. proceed to the red ball.
    6. navigate to the red ball.
- Open task
    1. open the green door.
    2. please open the green door.
    3. could you open the green door?
    4. unlock and open the green door.
    5. push the green door open.
    6. pull open the green door.
- Pickup task
    1. pick up the red ball.
    2. grab the red ball.

3. pick up the ball that is red.

4. retrieve the red ball.

5. lift the red ball.

6. take hold of the red ball.

- Put-next task

    1. put the red ball next to the blue key.

    2. place the red ball beside the blue key.

    3. move the red ball close to the blue key.

    4. set the red ball adjacent to the blue key.

    5. position the red ball near the blue key.

    6. arrange the red ball alongside the blue key.

**Rephrasing** We use 10 different natural language expressions as the novel goals generated by ChatGPT to express different target configuration. For example, if we take a goal configuration such as "red ball, blue key, green door", its corresponding natural language instructions can be one of the following:

- Goto task

    1. proceed in the vicinity of the red ball.

    2. move yourself toward the direction of the red ball.

- Open task

    1. leave the green door open.

    2. push the green door to open it fully.

    3. let the green door remain open.

    4. move aside the green door to open it.

    5. permit the green door to stay ajar.

    6. manipulate the green door into an open state.

- Pickup task

    1. grip the red ball.

    2. snag hold of the red ball.

    3. clasp the red ball.

    4. reach over and take the red ball.

    5. obtain and hold the red ball.

    6. gather the red ball into your hands.

- Put-next task

    1. position the red ball right alongside the blue key.

    2. ensure the red ball is closely placed beside the blue key.

    3. make the red ball sit immediately next to the blue key.

    4. arrange the red ball neatly beside the blue key.

    5. move the red ball so that it is perfectly adjacent to the blue key.

**Easy**

- Open-go task

    – open the door, then goto any object.

- Open-pick task

    – open the door, then pick up any object.

- Go-wall task

    – goto the side of the wall.

- Go-center task

– goto the center of the room.

**Hard**

- Open-lock task
    - pick up the key, then open the door.
- Put-line task
    - put the three items in a line.
- Put-pile task
    - gather the three items into a pile.

### D.4.4 LIBERO

**Training** We use 20 different natural language expressions as training goals generated by ChatGPT to express different target configuration. For example, if we take a goal configuration such as "alphabet_soup", its corresponding natural language instruction can be one of the following:

- Pick task
    1. Employ the gripper to seize the alphabet_soup.
    2. Utilize the gripper for grasping the alphabet_soup.
- Place task
    1. Transfer the alphabet_soup to the basket.
    2. Shift the alphabet_soup to the basket.
    3. Position the alphabet_soup to the basket.
    4. Move the alphabet_soup to the basket.
    5. Place the alphabet_soup to the basket.
    6. Relocate the alphabet_soup to the basket.

**Rephrasing** We use 10 different natural language expressions as novel goals generated by ChatGPT to express different target configuration. For example, if we take a goal configuration such as "alphabet_soup", its corresponding natural language instruction can be one of the following:

- Pick task
    1. Employ the gripper tool to clasp the alphabet_soup.
    2. Utilize the gripping mechanism to hold the alphabet_soup.
- Place task
    1. Transport the alphabet_soup to the basket.
    2. Insert the alphabet_soup into the basket.

**Easy** In easy task, the agent needs to complete some unseen manipulation tasks. For example, if we take a goal configuration such as "alphabet_soup, cream_cheese", its corresponding natural language instruction can be the following:

- Pick-and-place task
    - Employ the gripper to seize the alphabet_soup and transfer the alphabet_soup to the basket.
- Pcik-and-place-unseen task
    - Employ the gripper to seize the alphabet_soup and transfer the alphabet_soup to the cream_cheese.
- Reach task
    - Employ the gripper to get close to the alphabet_soup.

**Hard** We design 3 types of unseen and complex tasks: combination of two simple tasks (Sequential-pick-and-place), high-level language comprehension task (Pick-and-place-aside, Sequential-pick-and-place-all) and safe task (Pick-out). For example, if we take a goal configuration such as "alphabet_soup, cream_cheese" for combination of two easy tasks, "alphabet_soup, cream_cheese, salad_dressing" for high-level language comprehension task and "alphabet_soup" for safe task, its corresponding natural language instruction can be the following:

- Sequential-pick-and-place task
    - Employ the gripper to seize the alphabet_soup and transfer the alphabet_soup to the basket. Then employ the gripper to seize the cream_cheese and transfer the cream_cheese to the basket.
- Pick-and-place-aside task
    - Employ the gripper to seize the alphabet_soup and transfer the alphabet_soup to the other side.
- Sequential-pick-and-place-all task
    - Employ the gripper to seize something and transfer it to the basket one by one until the alphabet_soup, cream_cheese and salad_dressing are all in the basket.
- Pick-out task
    - The basket is on fire, employ the gripper to seize the alphabet_soup in the basket and transfer the alphabet_soup out of the basket.

### D.4.5 MuJoCo

**Training** We use 10 different natural language expressions as training goals generated by ChatGPT to express different target configuration. Natural language instruction can be one of the following:

- Jump-forward task
    1. Jump a step forward.
- Jump-backward task
    1. Jump a step backward.
    2. Jump a step back.
- Run-forward task
    1. Run forward.
    2. Run ahead.
- Run-backward task
    1. Run backward.
    2. Run back.

**Rephrasing** We use 10 different natural language expressions as novel goals generated by ChatGPT to express different target configuration. Natural language instruction can be one of the following:

- Jump-forward task
    1. Jump a step forth.
    2. Jump one step ahead.
- Jump-backward task
    1. Jump one step backward.
    2. Jump one step back.
- Run-forward task
    1. Speed forward.
    2. Speed ahead.
- Run-backward task

1. Speed backward.

2. Speed back.

**Easy** In easy task, the agent needs to complete some novel locomotion tasks. Natural language instruction can be one of the following:

- Run-forward-fast task
    - Move forward faster.
- Run-backward-fast task
    - Move backward faster.

**Hard** We design 2 types of unseen and complex tasks: combination of two simple run tasks (Run-forward-then-backward, Run-backward-then-forward) and high-level language comprehension task (Jump-in-place). Natural language instruction can be one of the following:

- Run-forward-then-backward task
    - Move forward and slow down. Move backward.
- Run-backward-then-forward task
    - Move backward and slow down. Move forward.
- Jump-in-place task
    - Jump in the original position.

## E  IMPLEMENTATION DETAILS

### E.1  INTRODUCTION TO THE CODEBASE

**ImagineBench** The ImagineBench codebase is a benchmark for evaluating reinforcement learning algorithms that train the policies using both real data and imaginary rollouts from LLMs. In ImagineBench codebase, we provide offline RL algorithms in `imagineBench/algo` directory, 5 environments for evaluation in `imagineBench/envs` directory, evalution method in `imagineBench/evaluations.py` and data processing method in `imagineBench/utils.py`.

**Dataset** After getting Metaworld environment using `imagine_bench.make()`, both real data and imaginary rollouts are available with `env.get_dataset()` function. Here is an example for getting Metaworld real and rephrase dataset:

```python
import imagine_bench

# Optional task_level: ['real', 'rephrase', 'easy', 'hard'].
env = imagine_bench.make('MetaWorld-v0', level='rephrase')
real_data, imaginary_rollout_rephrase = env.get_dataset(level="rephrase")

# Or you can use the dataset with other task levels.
env = imagine_bench.make('MetaWorld-v0', level='easy')
real_data, imaginary_rollout_easy = env.get_dataset(level="easy")
```

**Training** We provide an example for offline RL training with d3rlpy using MuJoCo environment and its rephrase dataset:

```python
import d3rlpy
import imagine_bench
from imagine_bench.utils import LlataEncoderFactory, make_d3rlpy_dataset
from imagine_bench.evaluations import CallBack
env = imagine_bench.make('MuJoCo-v0', level='rephrase')
env_eval = imagine_bench.make('MuJoCo-v0', level='rephrase')
```

```
7   real_data, imaginary_rollout_rephrase = env.get_dataset(level="
        rephrase")
8   dataset = make_d3rlpy_dataset(real_data,
        imaginary_rollout_rephrase)
9
10  agent = d3rlpy.algos.TD3PlusBCConfig(
11              actor_encoder_factory=LlataEncoderFactory(feature_size
                    =256, hidden_size=256),
12              critic_encoder_factory=LlataEncoderFactory(
                    feature_size=256, hidden_size=256),
13      ).create(device="cuda:0")
14
15  callback = CallBack()
16  callback.add_eval_env(env_dict={'rephrase': env_eval}, eval_num
        =10)
17
18  agent.fit(
19          dataset=dataset,
20          n_steps=500000,
21          experiment_name="mujoco",
22          epoch_callback=callback.EvalCallback,
23      )
```

**Reproducibility** Here is an example for reproduce our result on BabyAI environment using bc algorithm and rephrase dataset:

```
1   python imagine_bench/train.py --algo bc --env BabyAI-v0 --ds_type
        rephrase
```

### E.2 PROMPTS FOR LLM SUPERVISED FINE-TUNING

- Dynamics prediction: *You are an expert in identifying environmental dynamics change. Current state is [$s_t$], after executing action [$a_t$], we get next state: [ANSWER].*

- Rollout to goal translation: *Translate the state/action rollout to textual goal.\n Rollout:[ROLLOUT]\n Goal: [ANSWER].*

- Goal to rollout translation: *Translate the textual goal to state/action rollout.\n Goal:[G].\n Rollout: [ANSWER]*

Here, [ANSWER] is the content that LLM should generate.

## F  ADDITIONAL RESULTS AND ANALYSIS

### F.1  MORE EXAMPLES OF THE LLM-IMAGINARY ROLLOUTS

We present additional examples of the LLM-imaginary rollouts in in Fig. 8. The rendered figures show that while the imaginary rollouts can reflect the object manipulation for simple goals, the consistency between the rollouts and the goals reduces when the goal becomes more complicated. This results call for better usage of the real rollouts to fine-tune LLM to generate high-quality imaginary rollouts.

### F.2  OVERALL COMPARISON OF OFFLINE RL BASELINES

We present overall comparison of offline RL baselines in Tab. 14, as a reference for algorithm selection in future application.

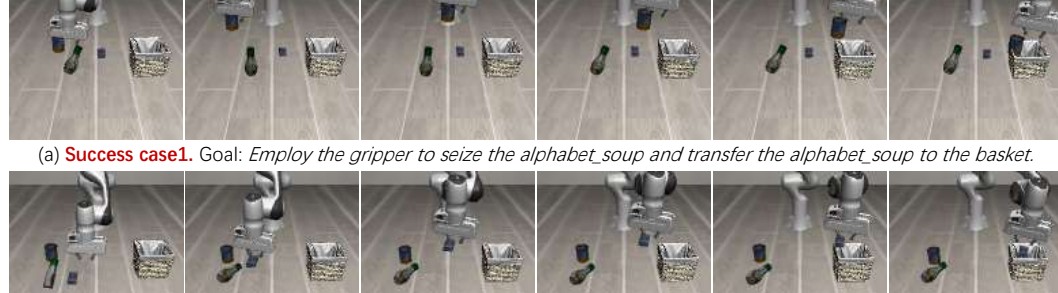

(a) **Success case1.** Goal: *Employ the gripper to seize the alphabet_soup and transfer the alphabet_soup to the basket.*

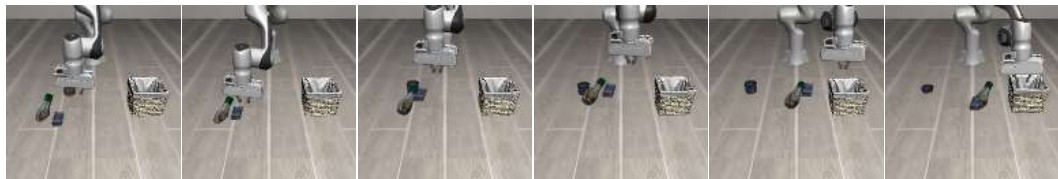

(b) **Success case2.** Goal: *Employ the gripper to seize the cream_cheese and transfer the cream_cheese to the basket.*

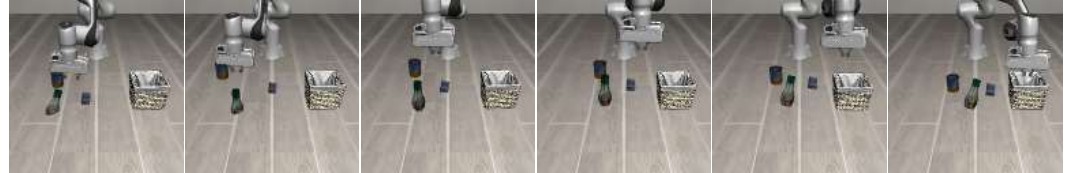

(c) **Failure case1.** Goal: *Employ the gripper to seize the salad_dressing and transfer the salad_dressing to the basket. Then employ the gripper to seize the cream_cheese and transfer the cream_cheese to the basket.*

(d) **Failure case2.** Goal: *Employ the gripper to seize something and transfer it to the basket one by one until the alphabet_soup, cream_cheese and salad_dressing are all in the basket.*

Figure 8: Examples of the LLM-imaginary rollouts for novel goals. The figures are obtained by rendering the states in LLM-imaginary rollouts.

### F.3 RESULTS WITH LLAMA-2-7B AS GENERATION MODEL

### F.4 TRAINING WITH DIFFERENT RATIOS OF IMAGINARY ROLLOUTS

We further investigate whether imaginary rollouts facilitate the acquisition of novel skills. To achieve this, we conduct ablation study on the ratios of the imaginary rollouts used for offline RL training, on BabyAI (rephrasing). As shown in Fig. 10, with larger amount of imaginary rollouts, different algorithms tend to get higher scores. This result serves as evidence that LLM-imaginary rollouts can effectively improve the performance on the novel tasks.

## G BROADER IMPACT STATEMENT

The development of RLIM holds potential for advancing adaptable and sample-efficient AI systems, with applications covering robotics, autonomous systems, and assistive technologies. By reducing reliance on costly real-world interaction data, RLIM could democratize access to advanced AI training, enabling smaller organizations and researchers to innovate in resource-constrained settings. The introduction of ImagineBench, an open-source benchmark, accelerates progress by standardizing evaluation across diverse tasks, from robotic manipulation to navigation. However, challenges such as computational costs from LLM fine-tuning and risks of synthetic data biases—which may propagate into deployed systems—warrant careful consideration. Ethical concerns around autonomous decision-making and environmental impacts of large-scale model training further underscore the need for responsible development. By addressing these challenges, RLIM could pave the way for safer, more generalizable AI agents capable of rapid adaptation in dynamic real-world environments, while its emphasis on instruction-following aligns with human-centric AI design, enhancing accessibility for non-expert users.

| Llama-2-7B | Train | Rephrase | Easy | Hard |
|---|---|---|---|---|
| **BC** | $65.96 \pm 15.07$ | $50.74 \pm 17.13$ | $35.22 \pm 29.60$ | $12.74 \pm 11.28$ |
| **BCQ** | $45.36 \pm 19.68$ | $42.08 \pm 17.34$ | $26.42 \pm 19.87$ | $11.56 \pm 14.00$ |
| **CQL** | $43.34 \pm 21.41$ | $37.28 \pm 20.88$ | $19.00 \pm 10.14$ | $11.74 \pm 11.43$ |
| **PRDC** | $42.57 \pm 27.97$ | $31.73 \pm 22.90$ | $31.20 \pm 36.97$ | $20.47 \pm 11.46$ |
| **TD3+BC** | $40.70 \pm 30.21$ | $28.03 \pm 19.17$ | $28.00 \pm 30.37$ | $16.83 \pm 13.06$ |
| **COMBO** | $27.87 \pm 29.44$ | $22.13 \pm 27.27$ | $21.77 \pm 24.78$ | $19.93 \pm 10.50$ |
| **SAC** | $5.40 \pm 2.10$ | $7.85 \pm 4.25$ | $16.70 \pm 8.30$ | $1.20 \pm 0.80$ |
| Qwen-3-4B | **Train** | **Rephrase** | **Easy** | **Hard** |
| **BC** | $67.78 \pm 16.64$ | $51.04 \pm 17.98$ | $25.96 \pm 8.70$ | $9.90 \pm 12.23$ |
| **BCQ** | $47.56 \pm 16.25$ | $43.04 \pm 16.03$ | $29.48 \pm 20.59$ | $8.62 \pm 10.15$ |
| **CQL** | $36.04 \pm 22.77$ | $35.78 \pm 22.38$ | $16.32 \pm 12.68$ | $9.64 \pm 12.20$ |
| **PRDC** | $40.40 \pm 29.81$ | $27.83 \pm 23.21$ | $32.07 \pm 40.05$ | $11.40 \pm 11.25$ |
| **TD3+BC** | $38.67 \pm 29.98$ | $30.47 \pm 27.15$ | $35.40 \pm 44.76$ | $14.27 \pm 13.97$ |
| **COMBO** | $45.20 \pm 29.60$ | $33.00 \pm 24.86$ | $16.87 \pm 11.54$ | $18.60 \pm 13.40$ |
| **SAC** | $5.25 \pm 0.45$ | $8.40 \pm 2.70$ | $16.70 \pm 8.30$ | $1.40 \pm 1.30$ |

Table 14: Overall comparison of offline RL baselines, with imaginary rollouts generated by Llama-2-7B (first table) and Qwen-3-4B (second table).

## H  USE OF LLMS

In this work, LLMs were used in two ways: (1) Pre-trained LLM (Qwen-3-4B-Instruct-2507 and Llama-2-7b-chat-hf) was fine-tuned on environment-collected rollouts to generate synthetic imaginary rollouts for novel tasks, as described in Section 4.2; (2) Publicly available LLM services were used for language polishing and grammatical refinement of the manuscript. The authors take full responsibility for all content, including the generated rollouts and the final text.

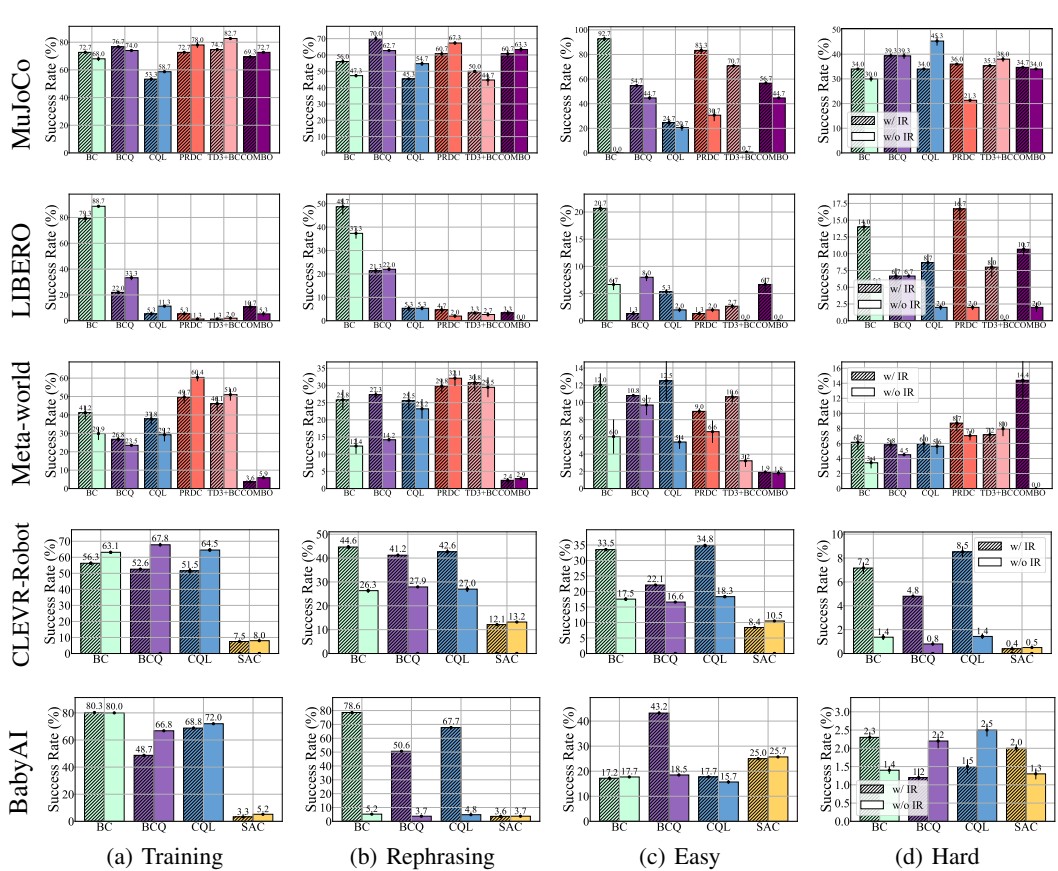

Figure 9: Success rate bars of different methods on various levels of goals, with imaginary rollouts generated by Llama-2-7B. The x-axis denotes the offline RL algorithm, and the y-axis denotes the success rate. 'w/ IR' stands for training with both real and imaginary rollouts. The success rate is averaged over the last five checkpoints, and the error bars are the half standard deviation over three seeds. We provide the results for Qwen-3-4B in Sec. 5.2.

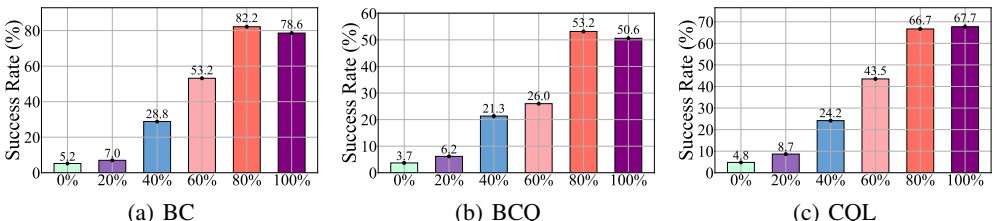

Figure 10: Success rate bars of different methods trained on various ratios imaginary rollouts. The x-axis denotes the ratio of used imaginary offline RL data, and the y-axis denotes the success rate for completing various natural language goals. The success rate is calculated based on the average of the last five checkpoints, and the error bars stand for the half standard deviation over three random seeds.

