# OpenReview forum: "ImagineBench: Evaluating Reinforcement Learning with Large Language Model Rollouts"
_ICLR.cc/2026/Conference — ICLR 2026 Conference Withdrawn Submission_

### Official Review · Reviewer_4QYi · 2025-10-21

**Soundness:** 3
**Presentation:** 2
**Contribution:** 3
**Rating:** 6
**Confidence:** 3

**Summary:**

This paper introduces ImagineBench, a benchmark for evaluating Reinforcement Learning from Large Language Model  Rollouts. The benchmark provides datasets containing both real environment rollouts and synthetic rollouts generated by fine-tuned LLMs (Qwen-3-4B and Llama-2-7B). ImagineBench spans diverse domains—locomotion, robotic manipulation, and navigation—and includes tasks with hierarchical difficulty levels (training, rephrasing, easy, hard). Using this benchmark, the authors evaluate several offline RL algorithms (e.g., BC, CQL, BCQ, TD3+BC, COMBO) trained with and without LLM-imaginary rollouts. They report that imaginary rollouts improve performance on new tasks. The paper also assesses the fidelity of LLM rollouts and shows that pretraining with these rollouts accelerates subsequent online adaptation.

**Strengths:**

1. Timely and relevant contribution —The paper addresses an emerging topic at the intersection of LLMs and reinforcement learning by providing a standardized benchmark for RL from imaginary LLM rollouts.

2. Comprehensive design — ImagineBench covers multiple environments (Meta-World, LIBERO, BabyAI, CLEVR-Robot, MuJoCo) and diverse goals with different levels of complexity.

3. Empirical evaluation — The study benchmarks a set of offline RL methods, providing valuable reference results for future work.

**Weaknesses:**

1. Methodological transparency in LLM fine-tuning — The paper briefly mentions adding layers to handle environmental data but lacks details on architecture modifications, representation of continuous variables, or training objectives.

2. Methodological transparency in Section 5 — some aspects of the evaluation in Section 5 are unclear (see Questions for more details).

3. Table 2 only reports aggregate success, transition, and legality scores for one environment. A more systematic analysis between rollout quality and downstream RL performance (e.g., by environment or goal type) would strengthen the claims.

4. Writing issues:
    - “How is the quality of the LLM-imaginary rollouts” → “What is the quality of …”
    - Lines 275-276 are ungrammatical

**Questions:**

1. What is the reasoning behind the choice of expert policies in real rollout collection?

2. In Sec. 4.2: “We use a pre-trained LLM as the backbone model and modify it with additional layers to handle environmental data.” Could the authors elaborate on these modifications—e.g., how are continuous vectors represented? Is the back-bone frozen or fine-tuned jointly?

3. Could the authors clarify what “unseen tasks” refer to in Section 5? From my understanding, all LLM-generated goals are used to produce imaginary rollouts for training the w/ IR baselines, so it’s unclear whether any tasks are actually held out for evaluation. Are some real or imaginary goals excluded from training and reserved for testing?

4. Could the authors compare the fidelity of LLM-generated rollouts (Table 2) with the downstream success rate of offline RL methods trained with/without IR, possibly in a consolidated table grouped by goal difficulty and environment (as in Fig. 4)? If the fine-tuned LLM is sufficiently good in generating a good state-action sequence, then the LLM itself can act as a goal conditioned policy (as in [1]). This comparison would help in understanding the benefit of the additional step of offline RL.

5. How many rollouts per goal are generated by the fine-tuned LLM, and how does this number influences the downstream policy performance?

6. How are "Real" rollouts for novel tasks obtained in section 5.3?

7. If each imaginary goal has an associated ground-truth reward (line 290), have the authors considered fine-tuning the LLM online using this reward (e.g., via GRPO or PPO) instead of the offline SFT on training goal rollouts?

---

References:

[1] https://arxiv.org/abs/2412.06877 — The Synergy of LLMs & RL Unlocks Offline Learning of Generalizable Language-Conditioned Policies with Low-fidelity Data

---

### Official Review · Reviewer_m12H · 2025-10-29

**Soundness:** 2
**Presentation:** 2
**Contribution:** 2
**Rating:** 2
**Confidence:** 5

**Summary:**

magineBench: Evaluating Reinforcement Learning with Large Language Model Rollouts introduces the standardized benchmark for Reinforcement Learning from Imaginary Rollouts (RLIM), a paradigm where large language models (LLMs) generate synthetic (“imaginary”) rollouts to augment real-world data for policy learning. ImagineBench provides curated datasets that pair real and LLM-generated rollouts across multiple domains, including locomotion, manipulation, and navigation, with natural language task instructions at varying difficulty levels. Experiments reveal that existing offline RL algorithms, when trained on both real and imaginary rollouts, achieve limited generalization on unseen tasks (35.44% vs. 64.37% for real rollouts), exposing the current gap between synthetic and real experience. Nonetheless, pretraining with imaginary rollouts improves downstream fine-tuning and accelerates online adaptation. The benchmark also quantifies the quality of LLM-generated trajectories and shows that larger LLMs produce more consistent, physically valid rollouts. Overall, ImagineBench provides a unified platform for evaluating how well RL algorithms leverage imagined experience, guiding future work on algorithmic design, continual adaptation, and multimodal integration.

**Strengths:**

The paper presents *ImagineBench*, a large-scale benchmark designed to evaluate reinforcement learning (RL) algorithms trained with both real and LLM-generated (“imaginary”) rollouts. The motivation is relevant, as the community lacks standardized evaluation for LLM-driven synthetic data in RL. The framework integrates multiple existing environments (Meta-World, LIBERO, MuJoCo, etc.) under a unified interface and provides detailed documentation, hierarchical task levels, and open-source code for reproducibility. The authors conduct comprehensive experiments across various offline RL algorithms, revealing consistent trends and challenges in leveraging imaginary data for policy generalization. The paper is clearly written, methodologically organized, and provides an informative dataset and evaluation pipeline that might serve as a useful resource for future studies on synthetic data-based RL benchmarking, even if the overall contribution remains primarily infrastructural rather than algorithmically novel.

**Weaknesses:**

The proposed benchmark raises several weaknesses:

1.This benchmark seems simply the integrating of existing simulation benchmarks (with Meta-world, LIBERO, Mujoco etc.). There is no new simulation scenarios are introduced. The massive integration and providing universal interface are already done by previous works such as RoboVerse.

2.The benchmark tasks that author selected are only focusing on relative simple manipulation tasks in which completion require only one primitive or locomotion tasks that with highly simplified robot dynamics. This benchmark lack of evaluation on data generation quality in facing more complex task scenarios such as long-horizon tasks and dexterous in hand manipulation.

3.The benchmark evaluate how LLM generated synthesis data can contribute to robotic RL learning. However, the author only finetuned two small size LLM (Qwen-3-4B and Llama-2-7B). Moreover, there is a lack of comparative analysis regarding how fine-tuning methods and LLM model size impact generation outcomes, resulting in insufficient persuasiveness due to ablation results gaps.

4.In the robot learning tasks, simply validate the generalization capability ONLY in the simulation is far not enough. In the real-world deployment, the agent may be interfered by a small noise (e.g. jitter, light deviation, sensor measurement error), and cause total failure of whole task, regardless how well they performed in the simulation. The synthetic data generated by LLMs proposed by the authors regrettably fails to account for these factors. Furthermore, the evaluation metrics for synthetic data contributions provided by the authors cannot demonstrate the extent to which LLM-generated synthetic data can bridge the sim2real gap , which is a critical problem in robotic tasks. Additionally, the selected toy environments (e.g., BabyAI) lack evaluative value and relevance in real-world contexts.

5.The robotic reinforcement learning is heavily rely on the reward signal to identify which state-action pair is good and which not. However, when handover the data acquisition to LLM generation, it is not guarantee that the generated action and state is aligned with their reward signal. And authors are not systematically analyze this aspect. The raise the concern about data quality generated by finetuned LLM. It is difficult to gain an intuitive understanding of this issue solely through success rates. Since this is a benchmarking paper rather than one introducing a technically innovative reinforcement learning framework for robots, I think the authors have an obligation to explicitly quantify the depth of their analysis regarding reward signals, environmental states, and action alignment issues (e.g., by providing visual results or novel evaluation metrics).

I find the arguments presented in this benchmark paper somewhat interesting, but its contribution falls short due to the lack of critical experimental evidence supporting the validity and effectiveness of its benchmark design. Key issues include (1) insufficient comparison of LLM capabilities, (2) contributions to sim2real gaps, and (3) concerns regarding reward alignment and configuration. Given the limited timeframe for a response, I doubt the authors can produce genuinely valid experimental results addressing these critical questions. Therefore, I recommend revising the paper by supplementing relevant results and updating its conclusions before resubmission to elsewhere.

**Questions:**

1.The authors claimed that in the first step they collect expert data from existing benchmarks. However, benchmarks such as LIBERO has already provided dataset for imitation learning. Did authors re-collect and extend dataset? The performance of LLM generation is also influenced by expert data. However, there is no explanation how well the expert agent perform in their individual tasks. The authors need to further provide detailed information related to this part in their appendix.

2.For the data filtering mechanism. Why authors not consider hindsight goal relabeling for their real data diversity? And how the author guarantee that the LLM generation can cover diverse task distributions?

3.It is unclear about the task level setup. Especially how to identify what kind of task can be regarded as easy and what kind of task is hard? In my opinion for all meta-world tasks can be classified as easy and libero tasks are complex as all meta-world tasks require only one primitive. By comparison, libero tasks on the one hand require more complex spatial understanding and on the other hand, some variations require multiple primitives to build long-horizon. But I thinks this is not the authors original intention. The author needs to further clarify their classification, especially with quantitative evidence.

4.The definition and calculation of metrics for Transition and Legality are unclear; the author needs to clarify how to calculate these values.

5.Again, there is no quantitative evidence or ablation study for comparative analysis regarding how fine-tuning methods and LLM model size impact generation outcomes.

---

### Official Review · Reviewer_Vz9R · 2025-10-31

**Soundness:** 2
**Presentation:** 3
**Contribution:** 1
**Rating:** 2
**Confidence:** 3

**Summary:**

This paper introduces a new benchmark for reinforcement learning. More precisely, it introduce ImagineBench, a benchmark that aims to study the interplay of RL training when coupled with LLM generated data. The work provides novel datasets generated by finetuned LLMs for several benchmarks. Then, it evalutes various baselines across these benchmarks.

**Strengths:**

**Strengths**
**Clarity**
* The paper is well written and uses clear language.
* All components of the benchmark are sufficiently well described for me to understand the tasks and objectives of the benchmark.

**Novelty**
* I’m not aware of any benchmark for studying the effect of LLM generated rollouts in RL.

**Experimental evaluation**
* There is a quite extensive set of experiments in this paper which is quite laudable.

**Weaknesses:**

**Clarity**
* The Figures are very difficult to read
* The analysis of the generated data should come before any RL experiments are executed. Otherwise, it is unclear why the RL agents perform the way they do.

**Related Work**
* The related work seems to largely focus on LLM generated data and it is unclear to me why generating data is specific to LLMs. To me, data generation with diffusion is just as relevant to the topic and there is plenty of work on this out there.

**Motivation**
* It is unclear to me why generated data in any benchmark would have to be limited to fine-tuned LLMs and why specifically the data provided in this benchmark is what should be used to study RL on generated data. It is unclear to me why relevant work on diffusion models does not do the same job but often better.
* The quality of the generated data hinges on the quality of the model that generates it. At some point in the later sections, the work states that the larger model (often known to have better performance on generation tasks in language) creates better generations. It seems that the topics of artificial generation and training from generated data are intertwined in that if I can train a better generator, I can get away with fewer policy learning adjustments. So it seems odd that one would I fix a set of datapoints from a model that is arguably not the best one that exists.
* I am also worried that depending on the pretraining of the LLM, each LLM might have different behavior when generating data for control tasks. It is well known that various LLMs available have their own peculiarities and it is unclear why that wouldn't translate to control. Thus, the choice of LLM might be crucial for development and understanding of all the general phenomena at play. Fixing one LLM seems imprudent.
* It is unclear to me why I would want or expect any RL algorithm to succeed on a task for which my data is only 60% or 80% of the time consistent with the MDP that I am trying to solve.

**Empirical Design, Claims and Evidence**
* I find it difficult to grasp the intended message of the benchmark results. The main takeaway appears to be that if the generative model produces poor-quality data, task performance naturally suffers. If that’s the case, the issue might simply be mitigated by improving the generative model itself, so it would help to clarify what deeper insight or challenge these results are meant to highlight.
* Relatedly, the purpose of a benchmark in general should be to study a specific phenomenon and conclude insights about it. It is unclear to me what the conclusions are from the results as many of the findings are intertwined with each other. For example, it is unclear when LLM generated data might help on unseen tasks, and whenever it does, why.
* The data quality should really be analyzed for all tasks not only Baby AI. It seems likely that generation difficulty hinges on environment complexity. A benchmark paper should be able to demonstrate what the properties of it’s benchmark or data are that is currently not the case for all tasks.
* Relatedly, there are no additional ablations that would tell the reader about the cross-correlation of the effects from generated data onto policy training.
* The previous point should come with a set of metrics that let me study whether I have removed an issue with my algorithm that is introduces by the generated data. However, the main metric seems to be success which limits generated insights.
* There are various claims that remain hypothetical and no evidence is provided. I believe it would make for a stronger paper if these claims were supported by evidence. For example:
    * L363: “BCQ and CQL achieve superior sample efficiency and stability in high-dimensional action spaces” - I don’t see which part of Figure 4 would support this claim.
    * L365: “There is clear performance degradation on hard tasks, with most methods’ success rates below 10% on Meta-World, CLEVR-Robot, and BabyAI. This gap could stem from the suboptimal reward function with current LLM rollouts,” - At this point there is no evidence for that provided. Why would it not be that the data that is generated is simply inconsistent with the state space for instance? That is one finding later in the paper.
    * L400: “The conservative learning nature of CQL allows it to focus on the state’s features, potentially enabling it to perform well on rephrasing even when using only real rollouts for training tasks.” - There is no analysis for this effect and this is a statement that is not supported.

**Questions:**

Q1: I find the distinction of the benchmark in the related work a bit confusing. It distinguishes the benchmark from offline RL benchmarks by saying "In contrast, ImagineBench is the first benchmark specifically designed to evaluate how effectively RL algorithms utilize LLM-imaginary rollouts”. But it is still an offline RL benchmark, am I missing something?

Q2: For the unseen task settings, is the generated data generated on the unseen task?

---

### Official Review · Reviewer_GnhX · 2025-11-01

**Soundness:** 3
**Presentation:** 1
**Contribution:** 2
**Rating:** 2
**Confidence:** 4

**Summary:**

This paper introduces a benchmark for assessing RL algorithms that leverage both real-world and LLM-generated "imaginary rollouts." The benchmark provides standardized datasets combining environment-collected and LLM-generated experiences across locomotion, manipulation, and navigation tasks, accompanied by natural language instructions of varying complexity. Based on the results, existing offline RL methods achieve limited generalization when trained on imaginary rollouts. It further shows that pretraining with imaginary rollouts can accelerate online adaptation.

**Strengths:**

The paper provides extensive experiments demonstrating the strengths and limitations of existing offline RL methods when trained on synthetic data.

**Weaknesses:**

1) Typo in the title: It should be "Rollouts" instead of "Rollout." It is correctly written in OpenReview but with a typo in the PDF. In addition with the white space issues throughout the paper (e.g., before section titles of after Figure 7), it feels like the paper was overlength and the authors tried to cut it way too aggressively.
2) Line 74: What is the difference between novel and unseen?
3) The related work section should also discuss methods that learn/use dynamics models (or as they are called now: world models), because the LLM is basically used as a dynamics model here to simulate dynamics of the real world.
4) The caption in Table 1 should say "rollouts" instead of "rollout."
5) While reading the paper, around line 257, I questioned "why does the paper stick with training prompts? Given that an LLM is available, why not add augmentations based on rephrasing? This would, in expectation, increase performance." I was thinking this also because lines 277-280 make it look like rephrasing is done only for evaluation datasets. But then, around line 422, the implication is that rephrasing is done during training as well. However, this brings the question: How about the rephrasing evaluations? Do those evaluation sets include prompts from the imaginary rollouts? The answers to these questions are not clear in the paper, which significantly hurts readability.
6) Figure 4 is not very readable, and requires significant polishing. For example, the numbers are too small and the grids are unnecessary.
7) In column d of Figure 4, the success rates are so low that I do not know if those numbers are meaningful for comparing different methods (except MuJoCo).
8) Line 416 has an incomplete sentence: "the states are ..."
9) The success rate metric measures the ratios of the imaginary rollouts that successfully complete the given goals. But how does the paper evaluate the success of infeasible trajectories? For example, what if the rollout takes mostly correct actions and reaches the goal state at the end, but somewhere in the middle, it teleports from a location to another which is not feasible in the real world? The metric needs clarification.
10) The paper shares the result that larger backbone LLM's generation quality clearly outperforms the small model as a "notable" result (lines 419-420). I do not understand why this is notable. Is this not the expected outcome? What should the reader note about this?
11) Section 6 finally brings back the acronym RLIM. This acronym was used in the Introduction but then completely abandoned until Section 6. While this itself is not a big concern, in general, the way the paper is written makes it feel like the authors tried to develop a method RLIM, but they later decided to frame the works as a benchmark instead of a new algorithm. I do not understand why it is a benchmark though. Truly, it is an algorithm. What is the reason of motivation behind trying to frame it as a benchmark? And even if we assume it is a benchmark, it is a benchmark for what? RL algorithms or LLMs? Why one and not the other?

**Questions:**

I asked several questions in the Weaknesses section, please see those questions.

---

### Note · Authors · 2025-12-20

**Comment:**

Dear Area Chair and Reviewers,

We will update the paper to incorporate the valuable feedback received during the review process. We sincerely thank the reviewers for their detailed and constructive comments.

Based on your reviews, we plan to improve the work in the following key directions:

- Clarify positioning: We will resolve the ambiguity between the proposed benchmark and the RLIM method to ensure the paper's contribution is framed more precisely.
- Deepen data analysis: We will conduct a more rigorous analysis of the quality of LLM-generated rollouts, including investigating the correlation between rollout fidelity (e.g., transition/legality) and downstream RL performance.
- More details about LLM fine-tuning: We will provide more details regarding the LLM fine-tuning process, specifically how continuous environmental data is handled.
- Presentation: We will correct the noted typos, polish the figures for readability, and ensure terminology (e.g., *unseen* vs. *novel*) is consistent throughout.

Thank you again for your time and effort in reviewing our work.

------

Best,

The Authors

**Withdrawal Confirmation:**

I have read and agree with the venue's withdrawal policy on behalf of myself and my co-authors.